# ASPD: Unlocking Adaptive Serial-Parallel Decoding by Exploring Intrinsic Parallelism in LLMs

## Abstract

The increasing scale and complexity of large language models (LLMs) pose significant inference latency challenges, primarily due to their autoregressive decoding paradigm characterized by the sequential nature of next-token prediction. By re-examining the outputs of autoregressive models, we observed that some segments exhibit parallelizable structures, which we term intrinsic parallelism. Decoding each parallelizable branch simultaneously (*i.e.* parallel decoding) can significantly improve the overall inference speed of LLMs. In this paper, we propose an **A**daptive **S**erial-**P**arallel **D**ecoding (**ASPD**), which addresses two core challenges: automated construction of parallelizable data and efficient parallel decoding mechanism. More specifically, we introduce a non-invasive pipeline that automatically extracts and validates parallelizable structures from the responses of autoregressive models. To empower efficient adaptive serial-parallel decoding, we implement a **Hybrid Decoding Engine** which enables seamless transitions between serial and parallel decoding modes while maintaining a reusable KV cache, maximizing computational efficiency. Extensive evaluations across General Tasks, Retrieval-Augmented Generation and Mathematical Reasoning demonstrate that **ASPD** achieves unprecedented performance in both effectiveness and efficiency. Notably, on Vicuna Bench, our method achieves up to 3.10x speedup (1.82x on average) while maintaining response quality within 1% difference compared to autoregressive models, realizing significant acceleration without compromising generation quality. The source code is available for review at an anonymous repository: https://anonymous.4open.science/r/ASPD.

## 1 Introduction

Recent advances in large language models (LLMs) (Yang et al., 2025a; Seed et al., 2025) have dramatically increased both model size and context length. While these improvements enhance model capabilities, they also increase inference latency due to the sequential nature of autoregressive decoding, posing challenges for practical applications that require low latency. Our analysis of LLM outputs reveals a key insight: despite being decoded sequentially, many model responses contain inherent parallelism that can be leveraged for parallel decoding. As illustrated in Figure 1, across various scenarios including general dialogue (LMSYS, 2023), our internal machine reading comprehension (MRC) benchmarks, Retrieval-Augmented Generation (Neural Bridge AI, 2023), and mathematical reasoning (Hugging Face, 2025), model responses consistently reveal significant potential for parallelization. By harnessing these inherently parallelizable segments for concurrent output, decoding speed can be substantially accelerated.

However, this also raises several fundamental technical challenges: First, identifying parallelizable segments while preserving semantic integrity is inherently difficult, due to the intricate dependencies present in natural language generation. Second, it is critical to ensure strict independence across parallel branches: each must remain contextually isolated during decoding, yet collectively yield a coherent output when merged. Third, the sophisticated coordination of positional information across parallel branches presents significant architectural challenges, particularly in preserving proper token relations and temporal coherence throughout parallel generation.

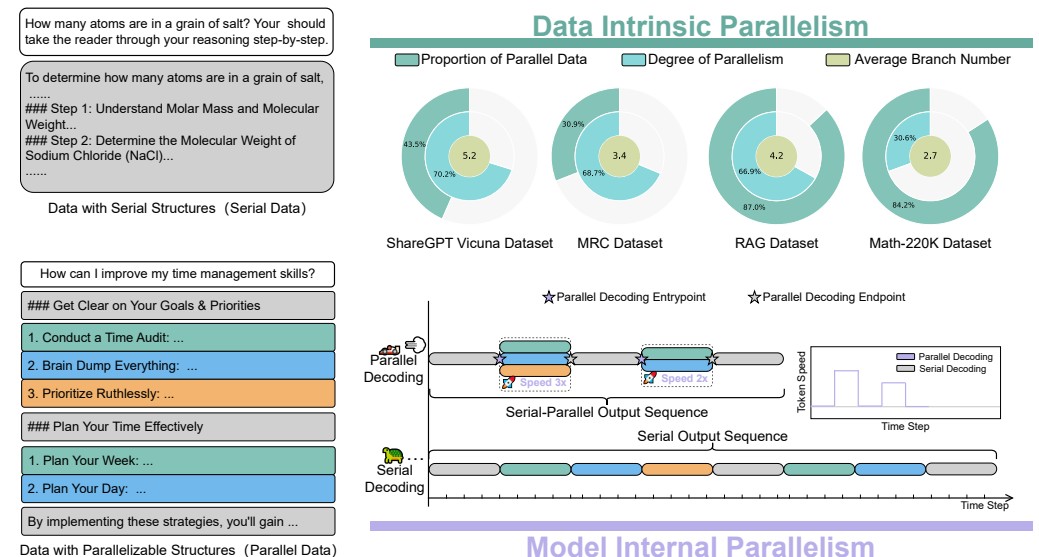

Figure 1: Overview of data intrinsic and model internal parallelism. The definition of Proportion of Parallel Data, Degree of Parallelism and Average Branch Number can be found in Section 4.1.

Recent attempts to address these challenges have yielded promising but limited solutions. APAR (Liu et al., 2024) discarded the KV-caches of parallel branches during generation to address position encoding issues, while compromising response quality. PASTA (Jin et al., 2025) explored asynchronous parallel decoding with pre-allocated position ranges, but struggles with position encoding mismatches of actual generation lengths.

Addressing these challenges is essential for unlocking the full potential of parallel decoding in practical LLM systems. To better exploit the intrinsic parallelism within autoregressive models, we present **A**daptive **S**erial-**P**arallel **D**ecoding, a novel framework that more effectively harnesses the model's inherent parallel capabilities through a dual-perspective optimization of data utilization and architectural innovation. Our approach first extracts inherent parallelism patterns from model responses to serve as training corpora for parallelization. We further propose an internal parallelization module that enables parallel processing in one go. By designing branch-specific attention masks and consistent positional ids across parallel branches, our method ensures that: (1) parallel branch generation maintains behavioral consistency with native serial decoding from each branch's perspective; and (2) upon completion of all parallel branches, switching back to the primary branch incurs no information loss and recomputation overhead. To achieve these objectives, we propose a Hybrid Decoding Engine that supports efficient parallel decoding and iterative serial-parallel decoding.

Our key contributions include:

- We develop an innovative non-invasive pipeline that automatically discovers and extracts inherent parallelizable structures from autoregressive model responses, which identifies semantically independent components that can be processed concurrently while preserving the response's original style. This enables us to build high-quality parallel training corpora automatically without altering the response probability distribution.

- We introduces a novel internal parallelization architecture that combines custom branch-invisible masking and branch-shared position embedding, enabling efficient parallel processing and seamless integration of parallel branches without batching or threading overhead. Based on this architecture, our proposed Hybrid Decoding Engine achieves efficient iterative serial-parallel decoding.

- Through comprehensive evaluation across diverse benchmarks - including general tasks (Vicuna Bench, MT Bench), retrieval-augmented generation (Neural-Bridge-RAG), and mathematical reasoning (MATH500, AMC23, GPQA, AIME2024, AIME2025) - we demonstrate significant improvements in both effectiveness and efficiency compared to existing approaches, achieving an optimal balance between these two critical metrics.

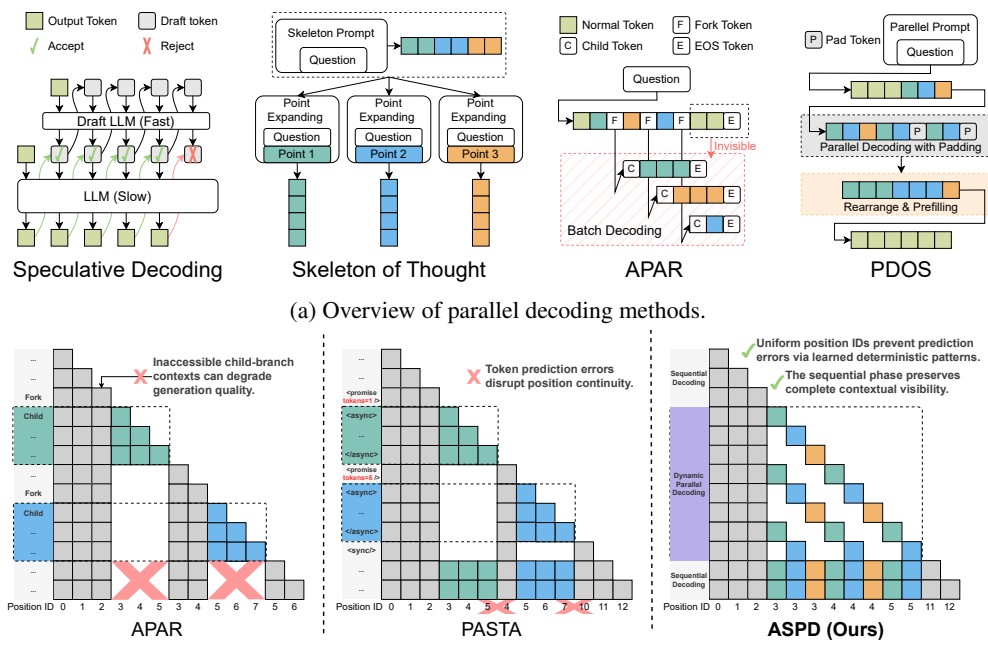

(a) Overview of parallel decoding methods.

(b) Internal parallel decoding mechanisms.

Figure 2: Related parallel decoding methods.

## 2 RELATED WORK

Existing research on parallelizing large language model inference can be categorized into two directions: enhancing response quality or accelerating generation speed. Quality-oriented methods (Brown et al., 2024; Wiseman & Rush, 2016; Wang et al., 2022; Chen et al., 2024; Zhang et al., 2024a; Yao et al., 2023) improve performance through multiple parallel sampling iterations but focus only on test-time scaling without optimizing sampling efficiency. Parallel scaling (Chen et al., 2025) enhances parallel computation in both training and inference by applying multiple learnable input transformations. In contrast, our work aligns with speed-oriented parallelization, which exploits inherent parallelism within a single response. This is achieved by decomposing sequential generation into parallelizable units while preserving textual coherence. A systematic taxonomy of this category is presented below.

**Orthogonal Acceleration Techniques** Speculative decoding, shown in Figure 2a, has recently emerged as a promising approach for accelerating LLM inference (Leviathan et al., 2023; Cai et al., 2023; Li et al., 2024a;b; 2025; Zhang et al., 2024b; Yi et al., 2024; Zhang et al., 2023). While effective, these techniques remain inherently sequential at the token level due to the autoregressive constraint, which fundamentally limits their achievable speedup.

**Parallelization via Prompt Engineering** This category exploits prompt engineering to facilitate parallel generation. As shown in Figure 2a, SoT (Ning et al., 2023) employs a two-phase outline-and-expand strategy, which introduces considerable overhead due to KV-cache reinitialization, batch processing, and restrictive prompt templates. PDOS (Yu, 2025) mitigates some of these issues through internal masks and logits processors, yet as a prompt-based approach, it still underutilizes the model's parallel capabilities and incurs efficiency loss from content re-prefilling during mode transitions.

**Architecture-Modified Parallelization** There have a few lines of work that modify attention mechanisms or training procedures to enable parallelism: (1) Visible Branch Architectures. Systems like GroupThink (Hsu et al., 2025) and Hogwild (Rodionov et al., 2025) allow inter-branch communication but suffer from backtracking costs (*e.g.*, when branches generate overlapping content). These methods focus more on branch collaboration rather than accelerating through independent parallel branch decoding. (2) Hidden Branch Architectures. As shown in Figure 2b, APAR (Liu et al., 2024),

PASTA (Jin et al., 2025), and APR (Pan et al., 2025) enforce strict branch isolation but inevitably face key limitations. APAR discards parallel branch KV-caches during integration, which undermines contextual coherence; APR only exchanges abbreviated summaries across modes rather than full KV-states, leading to degraded continuity; and PASTA relies on pre-allocated positional ranges, causing encoding conflicts when actual branch lengths diverge from predictions. Moreover, APAR depends on predefined rules, while PASTA lacks further validation of independence and completeness across parallel branches, which ultimately restricts their ability to exploit intrinsic parallelism.

Our concurrent work Multiverse (Yang et al., 2025b), focusing on the Mathematical Reasoning task, also explores parallelism in large language model generation, implementing parallel branch generation through SGLang (Zheng et al., 2024) and leveraging Radix Attention for KV cache reuse. In contrast, our approach performs decoding within a single sequence, providing inherent continuity of the KV cache to directly reuse.

The proposed **ASPD** framework better harnesses intrinsic parallelism through a non-invasive data transformation pipeline. By introducing branch-invisible masks and shared position encodings, we enable lossless and seamless transitions between serial and parallel decoding modes. We systematically investigate diverse model architectures and inference paradigms , demonstrating significant acceleration effects without quality degradation across multiple domains, particularly in general tasks and retrieval-augmented generation scenarios.

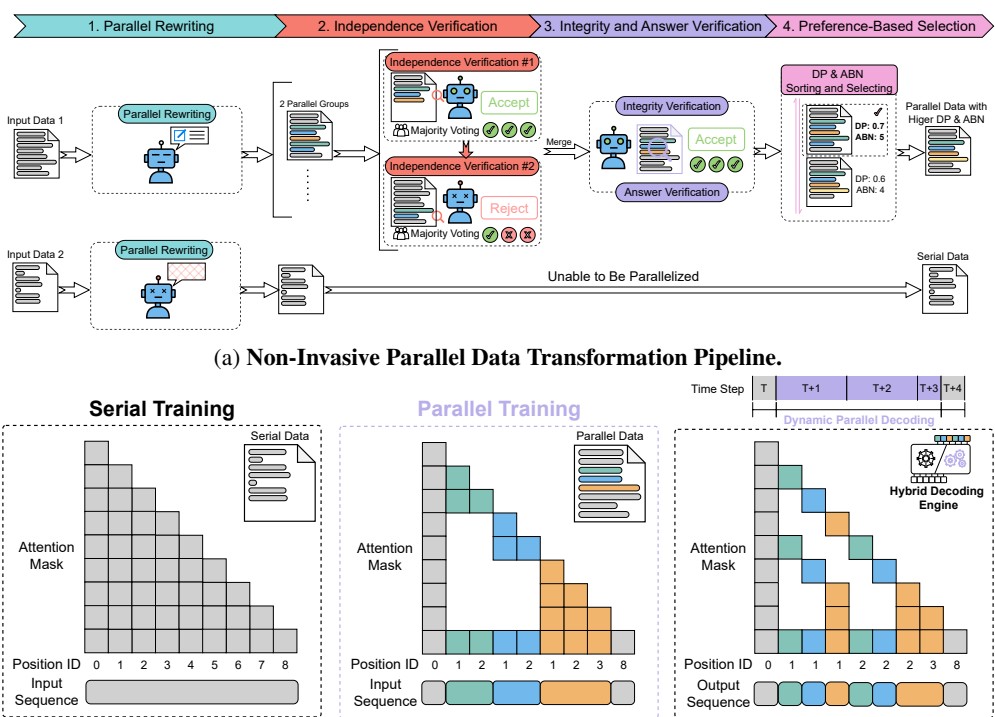

(a) **Non-Invasive Parallel Data Transformation Pipeline.**

(b) Serial-parallel training details (left-center). Hybrid Decoding Engine (right).

Figure 3: An overview of our **Adaptive Serial-Parallel Decoding** Framework.

## 3 METHODOLOGY

Our framework unlocks the intrinsic parallel capabilities of auto-regressive (AR) large language models through two key aspects: mining inherent parallelism in AR model responses and adapting internal model architectures. We achieve these objectives through three key components below:

## 3.1 Non-Invasive Parallel Data Transformation Pipeline

To effectively exploit this latent parallelism, we propose a novel non-invasive parallelization pipeline. Figure 3a illustrates the four foundational stages of our data curation pipeline. Given an original data sample $\langle Q, A \rangle$, where $Q$ is the question and $A$ is the model's response, we proceed as follows:

1. **Parallel Rewriting.** This step transforms the serial response $A$, which may contain implicit parallel structures, into a format with explicit parallel structure markup. This is achieved by feeding $A$ into the `Parallel Rewriting Prompt` and invoking an LLM. Specifically, multiple parallel branches (marked with the `<branch>` tag) are grouped into a parallel group (enclosed by the `<branchgroup>` tag), while serial content remains unchanged. The result is a rewritten answer, denoted as $A_{pr}$. The markup format is defined as:

   ```
   <branchgroup>
   <branch title="title 1">branch 1 content</branch>
   <branch title="title 2">branch 2 content</branch>
   ...
   </branchgroup>
   ```

   To promote diversity, this process is repeated $N$ times (with $N = 3$ by default), yielding a set of candidate rewritten answers: $\{A_{pr_1}, A_{pr_2}, \ldots, A_{pr_N}\}$.

2. **Independence Verification.** For each candidate $A_{pr_j}$ from the previous step, we iterate over every `<branchgroup>` to verify the mutual independence of its branches. The `Independence Verification Prompt` is populated with the original question $Q$, the content of all branches in the group, and any preceding serial text. The LLM is then prompted to assess whether the branches can be processed independently. If the majority of LLM judgments indicate dependence, the group **is reverted to serial form**. If all parallel groups in $A_{pr_j}$ fail the independence check, the candidate is discarded. Otherwise, the validated candidate—containing only verified parallel groups—is denoted as $A_{iv_j}$ and proceeds to the next stage.

3. **Integrity and Answer Verification.** For each surviving candidate $A_{iv_j}$, we first generate a temporary serialized version $A_{ia_j}$ by flattening all parallel groups back into sequential text. This $A_{ia_j}$ is then evaluated against the original answer $A$ using two separate prompts: the `Integrity Verification Prompt` checks for structural consistency, while the `Answer Verification Prompt` assesses answer correctness. A candidate $A_{iv_j}$ is retained only if both checks pass under majority voting. Candidates that fail either verification are discarded.

4. **Preference-Based Selection.** Among the $M$ candidates $\{A_{iv_1}, A_{iv_2}, \ldots, A_{iv_M}\}$ that pass all prior stages, we compute two metrics for each: the **Degree of Parallelism (DP)** and the **Average Branch Number (ABN)**. The candidate with the highest DP and ABN is selected as the final parallel-rewritten answer, denoted $A_{iv}$. The resulting parallel-structured data pair $\langle Q, A_{iv} \rangle$ is first transformed into the format supported by the subsequent Hybrid Decoding Engine, and is then included in the curated dataset.

If all candidates fail verification in Steps 2 or 3, the pipeline terminates for this sample, and the original data $\langle Q, A \rangle$ is preserved as the output. For each of the first three steps, input-output examples are provided in the corresponding prompts, all of which are shown in Appendix A.8.

## 3.2 Internal Parallelization Module

**Model Architecture for Native Parallelization**

After obtaining the parallelized corpus described in Section 3.1, we need to modify the model architecture to enable efficient serial-parallel decoding. As shown in Figure 3b, we need to handle both the visibility between branches and the position encoding within it. To better leverage the capabilities of the native autoregressive model during parallel phases while supporting seamless transitions between parallel and sequential modes, we introduce two key components: (1) an internal parallel

mask for branch-independent parallel decoding, and (2) shared positional encodings across parallel branches at the same timestamp.

**Preliminaries** The generation process comprises a sequence of interleaved stages, each decoding in serial or parallel mode. In serial stages, generation proceeds through a single main branch in an autoregressive manner. During parallel, the model simultaneously decodes multiple parallel branches, enabling concurrent token generation across different aspects. Here, $b(i)$ denotes the branch index of token $i$ and $t$ denotes temporal timestamps, $stage\_start(i)$ denotes the starting position of the stage where token $i$ is located, and $\Delta i$ represents the relative position of token $i$ within its stage. $P_t$ represents the number of tokens being decoded simultaneously at time $t$.

$$\text{Attn}(Q, K, V) = \text{softmax}\left(\frac{QK^\top}{\sqrt{d_k}} + M\right) V \tag{1}$$

$$M_{i,j} = \begin{cases} 0 & \text{if } \mathcal{S}(b(i), b(j)) = 1 \text{ and } \text{pos}(i) > \text{pos}(j) \\ -\infty & \text{otherwise} \end{cases} \tag{2}$$

$$\mathcal{S}(b(i), b(j)) = \begin{cases} 1 & \text{if } b(i) \text{ in main branch} \\ 1 & < b(i), b(j) > \text{ in same parallel branch} \\ 1 & < b(i), b(j) > \text{ in different stage} \\ 0 & \text{otherwise} \end{cases} \tag{3}$$

$$\text{pos}(i) = \begin{cases} \sum_{t=1}^{i-1} P_t + 1 & \text{if } i \text{ in main branch} \\ stage\_start(i) + \Delta i & \text{if } i \text{ in para branch} \end{cases} \tag{4}$$

We extend the vanilla causal attention (Eq. 1) by incorporating a visibility function $\mathcal{S}$. As shown in Eq. 2, $M_{i,j}$ is only visible when the visibility function $\mathcal{S}$ equals 1 and position j precedes position i. Eq. 3 defines the visibility between tokens, where the main branch can see all other branches, while for parallel branches, visibility is enabled when $< b(i), b(j) >$ spans across stages or belongs to the same parallel branch. For position encoding, as shown in Eq. 4, tokens in the main branch maintain absolute positions in the flattened sequence, while parallel branches synchronize their position encodings at each timestamp.

## 3.3 HYBRID DECODING ENGINE

To enable parallelization, we augment the vocabulary with six special tokens before training: `<title>`, `</title>`, `<branch>`, `</branch>`, `<para>`, and `</para>`. LLMs will learn how to use these tokens in the training phase. During inference, when the model determines that the current response can be parallelized, it generates several parallel titles, each enclosed by `<title>` and `</title>`. After all branch titles are generated, the model outputs a `<para>` token, turning the engine into parallel decoding mode. For each parallel branch $i$, we set the prefix "`<branch>`$T_i$: " to help the model identify which branch to generate, where $T_i$ is the corresponding title for branch $i$. A branch is considered complete when the `</branch>` token is reached. Once all branches are finished, a `</para>` token is appended to the output sequence, switching the engine back to serial decoding mode. Subsequently, the following decoding process iteratively repeats the aforementioned parallel and serial steps, achieving adaptive serial-parallel decoding. As shown in Figure 3b, the hybrid engine implements the branch-independent parallel mask and synchronized position-id mechanism proposed in Section 3.2 to support parallel decoding. Each parallel token attends only to tokens from both the main branch and its respective branch, with sequentially increasing position-ids. This allows each branch to maintain the same generation pattern as native autoregressive models from its own perspective. After all parallel branches complete decoding, sequential decoding resumes on the main branch, where the position id of the first token reflects its actual position in the complete sequence. Our hybrid decoding engine supports iterative serial-parallel switching for optimal efficiency.

## 4 EXPERIMENTS

### 4.1 EXPERIMENTAL SETUP

**Dataset** Following APAR (Liu et al., 2024), we use ShareGPT Vicuna dataset (LMSYS, 2023) as our training data. This dataset encompasses instructions across various scenarios including STEM, roleplay, reasoning, and extraction tasks. Additionally, we utilize our internal Machine Reading Comprehension (MRC) dataset for multi-dimensional validation (Corresponding experimental results can be found in Appendix A.5). All data processing is conducted using our proposed non-invasive parallelization transformation pipeline to obtain parallelized data. For fair comparison, we create corresponding sequential data by removing special parallel tokens.

**Evaluation** We conduct comprehensive evaluations using established benchmarks including Vicuna Bench (Chiang et al., 2023) and MT Bench (Zheng et al., 2023), following the evaluation protocol of APAR (Liu et al., 2024). Furthermore, to evaluate models' generalization capability , we introduce an out-of-domain benchmark, namely the RAG Bench. This benchmark consists of the first 200 questions with corresponding context sampled from rag-dataset-12000 (Neural Bridge AI, 2023).

Our evaluation framework encompasses two primary dimensions: **(1) Effectiveness Metrics.** We utilize the LLM-as-judge evaluation framework (Zheng et al., 2023) to quantify response quality, maintaining methodological consistency with APAR (Liu et al., 2024). All evaluations are conducted using Qwen3-235B-A22B (Yang et al., 2025a) to ensure balanced assessment. **(2) Efficiency Metrics.** We employ **T**okens-**P**er-**S**econd (**TPS**) as the primary throughput metric. For parallel models, we further introduce four additional metrics to characterize parallelization performance: **P**arallel-**T**okens-**P**er-**S**econd (**P-TPS**, TPS in parallel stage), **D**egree of **P**arallelism (**DP**, ratio of parallel to total tokens), **A**verage **B**ranch **N**um (**ABN**, average parallel branch num), **P**roportion of **P**arallel **D**ata (**PPD**).

Table 1: Performance comparison on MT Bench and Vicuna Bench. The **bold** and underlined values denote the best and second-best results, respectively.

| Bench \ Method | V-Ori | V-Seq | V-APAR | SoT | V-APAR* | **V-ASPD** | Q-Ori | Q-Seq | **Q-ASPD** |
|---|---|---|---|---|---|---|---|---|---|
| **MT Bench** | 4.86 | **5.59** | 4.88 | 4.48 | 5.38 | **5.59** | 7.82 | 7.98 | **8.15** |
| **Vicuna Bench** | 6.21 | 7.70 | 6.10 | 5.93 | 7.62 | **7.74** | 8.65 | **9.11** | 9.03 |

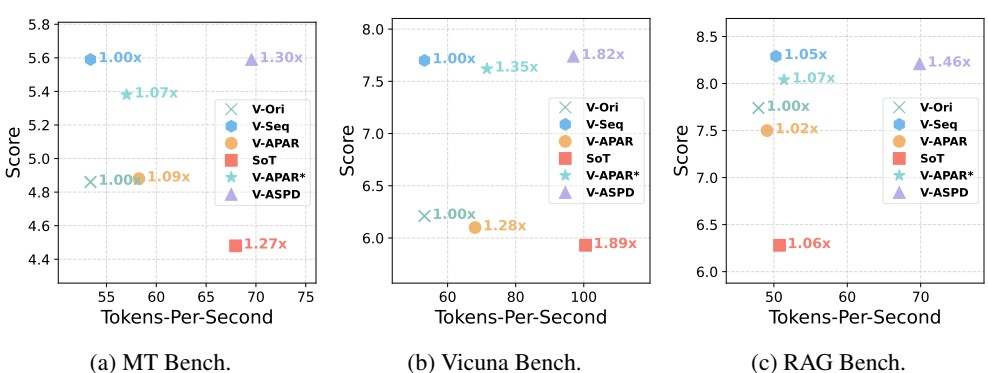

(a) MT Bench.     (b) Vicuna Bench.     (c) RAG Bench.

Figure 4: Speed-quality trade-off analysis across different parallel decoding methods on various benchmarks.

**Models** Following APAR (Liu et al., 2024), we employ Vicuna-V1.3-7B (Zheng et al., 2023) as our base model to ensure fair comparison across different approaches. To demonstrate the cross-architecture generalization capability of our method, we further conduct experiments using Qwen2.5-7b-Instruct (Qwen Team, 2024) as an additional foundation model.

**Implementation Details** During the training phase, we employed an initial learning rate of 1e-5 and a global batch size of 16 for three training epochs. The training process utilized cosine learning

rate scheduling with a warmup ratio of 0.1. The context length for training data was set to 8k tokens. Both sequential and parallel models used identical parameter configurations to ensure fair comparison. For inference, we use a batch size of 1, which aligns with common practices in parallel decoding literature, consistent with works such as PASTA(Jin et al., 2025) and Multiverse(Yang et al., 2025b). we consistently set the temperature parameter to 0.7, top_k to 20, and top_p to 0.8 across all experiments.

## 4.2 MAIN RESULTS

Tables 1 present a systematic evaluation across model architectures and methodologies. We evaluate our approach using two base models: Vicuna-1.3-7B (V) and Qwen2.5-7B-Instruct (Q). For each base model, we compare the original model (Ori), sequential fine-tuned model (Seq), SoT (Ning et al., 2023), APAR (Liu et al., 2024). Additionally, to ensure fair comparison, we utilize APAR's official codebase and enhance its training data quality using Qwen3-235B-A22B to obtain APAR*. The scores corresponding to each subtask are provided in Appendix A.2, and the efficiency metrics results are shown in Appendix A.2. The experimental results demonstrate the following:

**Superior Acceleration While Maintaining Quality** In Tables 1, both our fine-tuned parallel and serial models outperform the original model, establishing a solid foundation for our subsequent evaluations. Compared to the V-APAR and SoT method, our V-ASPD achieves a 14.55% and 24.78% improvement on the MT Bench, a 26.89% and 30.52% enhancement on the Vicuna Bench. In terms of decoding acceleration, presented in Figure 4, V-ASPD also achieves superior speed-quality balance across all benchmarks, with average acceleration ratios of 1.30x to 1.82x. Specifically, on Vicuna Bench, V-ASPD reaches 1.82x speedup, surpassing V-APAR (1.28x) and V-APAR* (1.35x), while nearly matching SoT's 1.89x that requires intensive memory usage. However, SoT's quality score of 5.93 is notably lower than both V-ASPD (7.74) and baseline (6.21), limited by its rigid prompt design. Furthermore, we observe speedup ratios ranging from 1.0x to 3.10x across Coding to Writing, outperforming V-APAR, V-APAR* and SoT on most subtasks (Detailed results are shown in Figure 6 in Appendix A.2). The consistent performance gains across diverse evaluation scenarios highlight the robustness and effectiveness of our parallel decoding framework.

**Cross-Domain and Cross-Model Generalization** As shown in Figure 4c, we observed that other competing methods face generalization issues on the out-of-domain RAG Bench. Especially, SoT's speedup drops to 1.06x compared to V-ASPD's 1.46x, due to requiring redundant prefilling of long context for each point during the point expanding stage. In contrast, our method not only achieves comparable generation quality to V-Seq but also maintains the highest speedup. The results presented in Table 1, which are based on the Qwen2.5-7B-Instruct model, highlight the consistent effectiveness of ASPD across model architectures. On MT Bench, Q-ASPD achieves a remarkable score of 8.15, demonstrating superior performance by surpassing both Q-Ori and Q-Seq models, with a notable 2.1% improvement compared to Q-Seq. Vicuna Bench shows comparable performance between Q-ASPD (9.03) and Q-Seq (9.11), with a mere 0.9% difference, confirming ASPD's output quality preservation. The minimal performance variation across benchmarks underscores ASPD's robustness.

## 4.3 PARALLELISM AT THE REASONING FRONTIER.

While previous approaches like APAR excluded mathematical and coding tasks from parallel data processing, we observe that our concurrent research work Multiverse (Yang et al., 2025b) has also directed attention toward mathematical problem parallelization. To thoroughly investigate this domain, we extended our model architecture to Qwen2.5-32B-Instruct (Qwen Team, 2024) and conducted comprehensive validation on these complex reasoning domains. Our evaluation includes a diverse range of benchmarks, including GPQA (Rein et al., 2024), MATH500 (Hendrycks et al., 2021), and competition-level mathematics such as AMC23 (MAA, 2023), AIME24 (MAA, 2024), and AIME25 (MAA, 2025). For training data, we utilize OpenR1-Math-220K (Hugging Face, 2025), which was generated by prompting DeepSeek-R1 (Guo et al., 2025) with NuminaMath1.5 (LI et al., 2024) as the query set. The model was trained for 9 epochs with a global batch size of 88, using a 12k context window and a learning rate of 1e-5.

As demonstrated in Table 2 (reporting pass@1 scores based on Evalchemy (Raoof et al., 2025),with AMC and AIME results representing means across 8 random seeds), ASPD exhibits performance

Table 2: Math bench performance with Qwen2.5-32B-Intruct.

| Bench \ Model | Ori | Seq | ASPD |
|---|---|---|---|
| **MATH500** | 82.00 | **94.40** | 94.00 |
| **AMC23** | 62.19 | **89.69** | 89.38 |
| **GPQA** | 48.99 | 61.11 | **65.66** |
| **AIME2024** | 17.50 | 58.75 | **62.08** |
| **AIME2025** | 12.50 | 47.92 | **50.00** |

Table 3: Comparative analysis of parallelization metrics on mathematical benchmarks between **ASPD** and **Seq.**

| Bench \ Metric | PPD | DP | ABN | TPS | P-TPS |
|---|---|---|---|---|---|
| **MATH500** | 88.40 | 33.30 | 2.61 | $27.14_{1.17x}$ | $43.03_{1.86x}$ |
| **AMC23** | 84.38 | 22.24 | 2.80 | $21.93_{1.11x}$ | $39.30_{1.99x}$ |
| **GPQA** | 66.16 | 32.70 | 2.88 | $22.06_{1.13x}$ | $36.57_{1.88x}$ |
| **AIME2024** | 65.42 | 8.84 | 2.48 | $16.43_{1.04x}$ | $24.37_{1.54x}$ |
| **AIME2025** | 79.17 | 8.60 | 2.40 | $15.77_{1.08x}$ | $26.82_{1.83x}$ |

gains of 12%, 27.19%, 16.67%, 44.58%, 37.5% over **Ori** across MATH500, AMC23, GPQA, AIME24 and AIME25 respectively. Notably, ASPD achieves acceleration ratios of 1.04-1.17 in TPS and 1.54-1.99 in P-TPS versus the **Seq** model, while maintaining performance within a range of -0.4% to +5%, demonstrating robust effectiveness.

## 4.4 ABLATION STUDY

### 4.4.1 IMPACT OF DATA PROCESSING PIPELINE.

While autoregressive models inherently possess parallelizable characteristics in their responses, precisely identifying parallelizable components remains a significant challenge. Ablation of data pipeline in Table 4 demonstrates the performance variations across different data processing methodologies. APAR achieves only 1.11x speedup compared to the baseline, due to its rule-based parallelization approach which fails to trigger parallel responses in many scenarios. PASTA exhibits the lowest performance score of 4.98, primarily because its data processing pipeline lacks consideration for branch independence verification. In contrast, our method achieves optimal results in both effectiveness (score of 7.64) and efficiency (TPS of 104.21), demonstrating that our Non-Invasive data transformation pipeline can better leverage the model's inherent parallel capabilities while maintaining superior response quality.

Table 4: Ablation studies on data pipeline efficiency, attention mask strategies, and position encoding schemes for parallel decoding. * denotes implementation with official codebase, † denotes implementation with official prompt.

| **Data Pipeline** | | | **Attention Mask** | | | | **Position Id** | | |
|---|---|---|---|---|---|---|---|---|---|
| **Method** | **Score** | **TPS** | **PosId** | **Attn** | **Score** | **TPS** | **PosId** | **Score** | **TPS** |
| Baseline | 6.21 | 53.19 | Seq | Shared | 4.64 | **110.30** | Predict | 6.75 | 72.15 |
| APAR* | 5.81 | 59.25 | **Seq** | **Indep** | **7.64** | 104.21 | Same-Max | 6.78 | 89.45 |
| PASTA† | 4.98 | **106.83** | Max | Shared | 3.70 | 86.96 | Same-Re | 7.29 | 95.24 |
| **ASPD** | **7.64** | 104.21 | Max | Indep | 6.78 | 89.45 | **Same-Seq** | **7.64** | **104.21** |

### 4.4.2 IMPACT OF MASK VISIBILITY

While GroupThink (Hsu et al., 2025) and Hogwild (Rodionov et al., 2025) have explored visible masks for agent collaboration, our approach differs fundamentally in its objective. As demonstrated in attention mask ablation of Table 4, we define two mask types: **Shared** and **Indep** (Independent), where branches can or cannot see each other respectively. We also introduce two types of position id setting: **Max** (uses maximum position id across parallel branches for merging) and **Seq** (assigns actual sequential position id to the first token of the merged main branch, as illustrated in Figure 3b). Our empirical evaluation shows that *Shared* masks consistently outperform *Indep* masks across both *Seq* and *Max* position id configurations. This architectural choice is motivated by our primary objective of decomposing responses into independent, self-contained parallel branches to achieve optimal computational efficiency. This empirical finding strongly validates our design decision to maintain strict branch isolation as an optimal strategy for parallel response generation.

### 4.4.3 Position Encoding Paradigms

During parallel decoding, multiple branches decode simultaneously while remaining mutually invisible, making the arrangement of position ids particularly crucial. Based on whether position ids are consistent across parallel branches, we abstract position encoding schemes into the following two categories: (1) **Predict.** Following PASTA (Jin et al., 2025), we allocate position ids to each parallel branch by predicting its length. Before parallel decoding starts, the model predict the prospective length of every branch and assigns initial position ids accordingly. We adopt PASTA's best-performing configuration, *Predict-10X*, which multiplies the predicted length by ten to set the decoding horizon for each branch. (2) **Same.** The following three variants use identical position ids across parallel branches at each timestamp, differing in their strategies for reordering position ids when merging parallel branches into the main branch. *Same-Max* equals to **Max** in Section 4.4.2. *Same-Re* (Same-Rearrange) reorders position ids sequentially between parallel branches. *Same-Seq* (Same-Sequential) equals to **Seq** in Section 4.4.2.

As evidenced in position id ablation of Table 4, due to inconsistencies between actual branch lengths during decoding and predicted lengths, the *Predict* strategy yields the poorest performance. Among the *Same* family approaches, our adopted *Same-Seq* strategy emerges as the most effective strategy, achieving a Score of 7.64 and TPS of 104.21, outperforming both *Predict*, *Same-Re* and *Same-Max* variants in terms of quality and efficiency. This suggests that maintaining natural position ordering while sharing timestamps across parallel branches optimally preserves the model's positional understanding.

## 5 Conclusion

In this work, we present **ASPD**: The **A**daptive **S**erial-**P**aralle **D**ecoding framework for efficient hybrid decoding within auto regressive large language models. The proposed method introduces a non-invasive parallel data transformation pipeline and internal parallelization with branch-invisible attention masks and shared position ids, achieving substantial latency reduction while maintaining response quality compared to traditional autoregressive LLMs. Our extensive experiments demonstrate state-of-the-art performance across various benchmarks including general tasks, retrieval-augmented generation, and mathematical reasoning. Furthermore, we establish a novel paradigm for parallel decoding that eliminates external overheads from batching, threading or re-prefill between serial-parallel switching, providing valuable insights for future research in efficient LLM inference. The strong empirical results and theoretical contributions of ASPD suggest promising applications in latency-critical scenarios.

## Reproducibility Statement

All experimental results presented in this work are fully reproducible. We provide comprehensive experimental settings in Section 4.1, including detailed implementation specifications covering datasets, evaluation protocols, and model architectures. To facilitate complete reproducibility of our research findings, we have made available a comprehensive anonymous code repository (referenced in the abstract) containing three essential components:

1. A parallel data construction pipeline (located in the `data_ppl` directory), which implements our Non-Invasive Parallel Data Transformation Pipeline.

2. A training framework implemented for the Internal Parallelization Module (located in the `train` directory), supporting for serial-parallel training.

3. An adaptive hybrid inference engine (located in the `infer` directory), which enables efficient serial-parallel decoding.

Each component is accompanied by thorough documentation and step-by-step operational guidelines to ensure complete reproducibility of our results. In accordance with the double-blind review process, all materials have been carefully anonymized to maintain review integrity.

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

## A    APPENDIX

### A.1    THE USE OF LARGE LANGUAGE MODELS (LLMS)

During the drafting of this work, AI tools were employed solely for two purposes: (1) conducting grammar and syntax checks to ensure linguistic accuracy and consistency; (2) polishing partial descriptive content and result presentation. Notably, the core content of this work—including research design, methodology, analytical frameworks, key parameters, and original conclusions—was independently conceived, developed, and finalized by the authors. AI tools did not participate in any substantive content creation, decision-making, or scientific reasoning processes. All AI-assisted revisions were further reviewed, verified, and adjusted by the authors to guarantee the academic rigor, authenticity, and integrity of the research content.

### A.2    FULL PERFORMANCE COMPARISON ON MT BENCH AND VICUNA BENCH

Table 6 shows complete experimental results of various subtasks on MT Bench and Vicuna Bench, and the corresponding acceleration ratio of each subtask is presented in Figure 6.

### A.3    WALL CLOCK LATENCY SPEEDUP

To evaluate the reduction in actual system overheads achieved by our method, we perform wall clock latency (WCL) speedup evaluations across benchmarks, and the results are consistent with the tokens/sec metrics we previously reported. As shown in Table 5, ASPD achieves the highest WCL speedup (44.90% on MT Bench, 33.90% on Vicuna Bench, and 37.64% on RAG Bench) compared to the sequential baseline (V-Seq) and the parallel baseline (V-APAR*). Notably, ASPD maintains generation quality comparable to V-Seq (e.g., 5.59 vs. 5.59 on MT Bench) while delivering superior

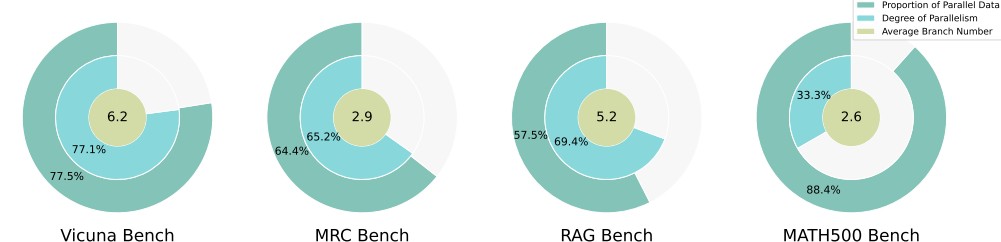

Figure 5: Parallelization patterns in our trained parallel model's responses across different scenarios.

Table 5: Comparison of WCL (wall clock latency) speedup and generation quality. * denotes implementation with official codebase.

| Model | MT Bench | | Vicuna Bench | | RAG Bench | |
|---|---|---|---|---|---|---|
| | WCL (Speedup) | Score | WCL (Speedup) | Score | WCL (Speedup) | Score |
| V-Seq | 15.5 | 5.59 | 17.64 | 7.70 | 9.67 | 8.29 |
| V-APAR* | 10.37 (33.10%) | 5.38 | 12.42 (29.59%) | 7.62 | 7.41 (23.37%) | 8.04 |
| V-ASPD | 8.54 (44.90%) | 5.59 | 11.66 (33.90%) | 7.74 | 6.03 (37.64%) | 8.21 |

speedup—further confirming the effectiveness of our method in balancing latency reduction and output quality.

### A.4  DATA INTRINSIC PARALLELISM OF BENCHMARKS

Building on our analysis of parallelization potential (Figure 1), our method effectively leverages intrinsic parallelism while maintaining generalizability across diverse scenarios (Figure 5). Results on Vicuna (Chiang et al., 2023), our Internal MRC, and RAG benchmark demonstrate optimal parallelism utilization. The Mathematical Reasoning (MATH500) (Hendrycks et al., 2021) benchmark exhibits the highest parallelization constraints among evaluated tasks, with 88.4% PPD (+4.2%), 33.3% DP (+2.7%), and 2.6 ABN (-0.1), which exhibits strong alignment with the statistical patterns shown in Figure 1. These quantitative metrics provide compelling evidence that **our approach successfully harnesses the inherent parallelization capabilities of LLMs while preserving the fundamental characteristics of native autoregressive generation.**

### A.5  GENERALIZATION ACROSS DIFFERENT BENCHMARKS

To evaluate the generalization capability of parallel processing across different benchmarks, we conducted training on our internal MRC benchmark and tested generalization performance on MRC, LR, and AI Search benchmarks, using Qwen2.5-7B-Instruct as our base model. Specifically, MRC is designed to evaluate a wide range of reading comprehension skills, including various question types such as single passage understanding, multi-information extraction, refusal to answer, etc. Logical Reasoning (LR) contains problems requiring step-by-step deductive reasoning, while AI Search resembles RAG scenarios where responses are generated by analyzing multi-source search results, typically presented in an itemized format. For these benchmarks, we implement human evaluation protocols to mitigate potential model-based assessment biases.

As shown in Table 8, on the homogeneous MRC benchmark, ASPD achieves a 1.35x speedup in TPS compared to the original model, while maintaining performance within 1% of the fine-tuned sequential model. Cross-validation using LR and AI Search benchmarks demonstrates acceleration ratios of 1.15x and 1.45x respectively.

For the AI Search test set, which contains more questions requiring itemized responses, we observe increased opportunities for parallel output generation. This leads to improved parallelization metrics including PPD, DP, and ABN (3.21 in AI Search versus 2.40 in Logical Reasoning), as illustrated in Table 7. Consequently, both TPS and P-TPS metrics show significant improvements. These results indicate that the effectiveness of our parallelization approach varies depending on the specific task characteristics.

Table 6: Performance comparison on MT Bench and Vicuna Bench. CS is short for Common-Sense on Vicuna Bench.

| MT Bench | | | | | | | | | |
|---|---|---|---|---|---|---|---|---|---|
| Model / Task | V-Ori | V-Seq | V-APAR | SoT | V-APAR* | **V-ASPD** | Q-Ori | Q-Seq | **Q-ASPD** |
| **Coding** | **3.50** | 3.10 | 2.70 | 3.00 | 2.55 | 2.90 | **7.15** | 6.60 | 6.70 |
| **Extraction** | 5.10 | 4.40 | **5.20** | 3.08 | 5.05 | 4.80 | 7.45 | 6.93 | **7.80** |
| **Humanities** | 6.55 | **8.40** | 6.40 | 6.30 | 8.10 | 7.95 | 8.70 | **9.15** | 9.05 |
| **Math** | 2.95 | **3.05** | 2.55 | 2.75 | 2.50 | 3.00 | 8.50 | 8.45 | **8.90** |
| **Reasoning** | 4.65 | 4.80 | 5.45 | 4.50 | 4.75 | **5.85** | 6.85 | **7.25** | 7.05 |
| **Roleplay** | 5.30 | 7.10 | 5.60 | 5.60 | 6.80 | **7.25** | 8.10 | 8.40 | **8.48** |
| **Stem** | 5.55 | 6.20 | 5.55 | 5.85 | **6.50** | 6.30 | 7.80 | **8.60** | 8.50 |
| **Writing** | 5.30 | **7.65** | 5.60 | 4.75 | 6.80 | 6.70 | 8.00 | 8.50 | **8.75** |
| **Mean** | 4.86 | **5.59** | 4.88 | 4.48 | 5.38 | 5.59 | 7.82 | 7.98 | **8.15** |
| **Vicuna Bench** | | | | | | | | | |
| Model / Task | V-Ori | V-Seq | V-APAR | SoT | V-APAR* | **V-ASPD** | Q-Ori | Q-Seq | **Q-ASPD** |
| **Coding** | **4.29** | 3.71 | 4.14 | 3.43 | 3.71 | 3.71 | **9.00** | 8.29 | 8.71 |
| **CS** | 7.40 | **9.00** | 7.20 | 7.30 | 8.90 | 8.90 | 9.00 | **9.30** | 9.00 |
| **Counterfactual** | 5.10 | 8.50 | 5.30 | 5.30 | 8.40 | **8.60** | 8.40 | **9.10** | 9.10 |
| **Fermi** | **5.50** | 4.50 | 4.60 | 4.40 | 5.00 | 5.20 | 7.70 | **8.20** | 8.10 |
| **Generic** | 7.30 | **9.20** | 7.40 | 7.40 | 8.95 | 8.80 | 8.70 | **9.55** | 9.40 |
| **Knowledge** | 7.30 | **9.10** | 7.20 | 7.50 | 8.60 | 9.00 | 9.00 | **9.40** | 9.10 |
| **Math** | 2.67 | 2.33 | **3.33** | 2.00 | 2.33 | 2.67 | 9.67 | **10.00** | **10.00** |
| **Roleplay** | 7.20 | **9.20** | 6.60 | 6.70 | 9.00 | 9.00 | 9.00 | 9.30 | **9.40** |
| **Writing** | 6.10 | 8.80 | 6.60 | 5.80 | 8.80 | **9.00** | 8.20 | **9.20** | 9.00 |
| **Mean** | 6.21 | 7.70 | 6.10 | 5.93 | 7.62 | **7.74** | 8.65 | **9.11** | 9.03 |

Table 7: Comprehensive parallelization metrics on our internal test sets.

| Test Set | PPD | DP | ABN | P-TPS |
|---|---|---|---|---|
| MRC | 64.40 | 65.21 | 2.86 | $109.32_{2.18x}$ |
| LR | 44.59 | 55.80 | 2.40 | $101.72_{1.92x}$ |
| AI Search | 64.50 | 67.69 | 3.21 | $97.27_{2.41x}$ |

Table 8: Human evaluation on our internal test sets.

| Model | MRC | | LR | | AI Search | |
|---|---|---|---|---|---|---|
| | ACC | TPS | ACC | TPS | Score | TPS |
| Q-Ori | 64.40 | 50.18 | 53.50 | 53.06 | 7.85 | 40.29 |
| Q-Seq | **73.20** | 50.53 | **56.69** | 54.32 | **8.23** | 39.89 |
| **Q-ASPD** | 72.40 | **67.91** | 55.41 | **61.26** | 8.15 | **58.56** |

## A.6 In-depth Analysis of Mathematical Results

**Parallelism Variation Across Tasks** Compared to the results of other scenarios in Figure 4, mathematical tasks demonstrate relatively lower acceleration benefits (1.17x speedup in MATH500, 1.46x in RAG, and 1.82x in Vicuna). In mathematical reasoning tasks, our parallel model contains reasoning processes with step-by-step deductions. These reasoning chains and strong inter-step dependencies lead to reduced DP, which aligns with the parallel pattern shown in Figure 1 (30.6% for mathematics versus approximately 68% for other domains). This reduction in parallelizable content consequently results in decreased TPS acceleration ratios.

**Parallelism Variation Across Difficulty Levels** As shown in Table 3, the DP significantly decreases from 33.30% in MATH500 to 8.6% in AIME2025, with TPS decreasing from 1.17x to 1.08x. This decline in parallelism indicates a strong correlation between performance degradation and difficulty level. Our detailed analysis of MATH500 in Figure 7a clearly shows that parallel efficiency metrics decrease as problem difficulty increases. This inverse relationship is consistently observed across various mathematical benchmarks in Figure 7b, where higher difficulty levels corresponds to

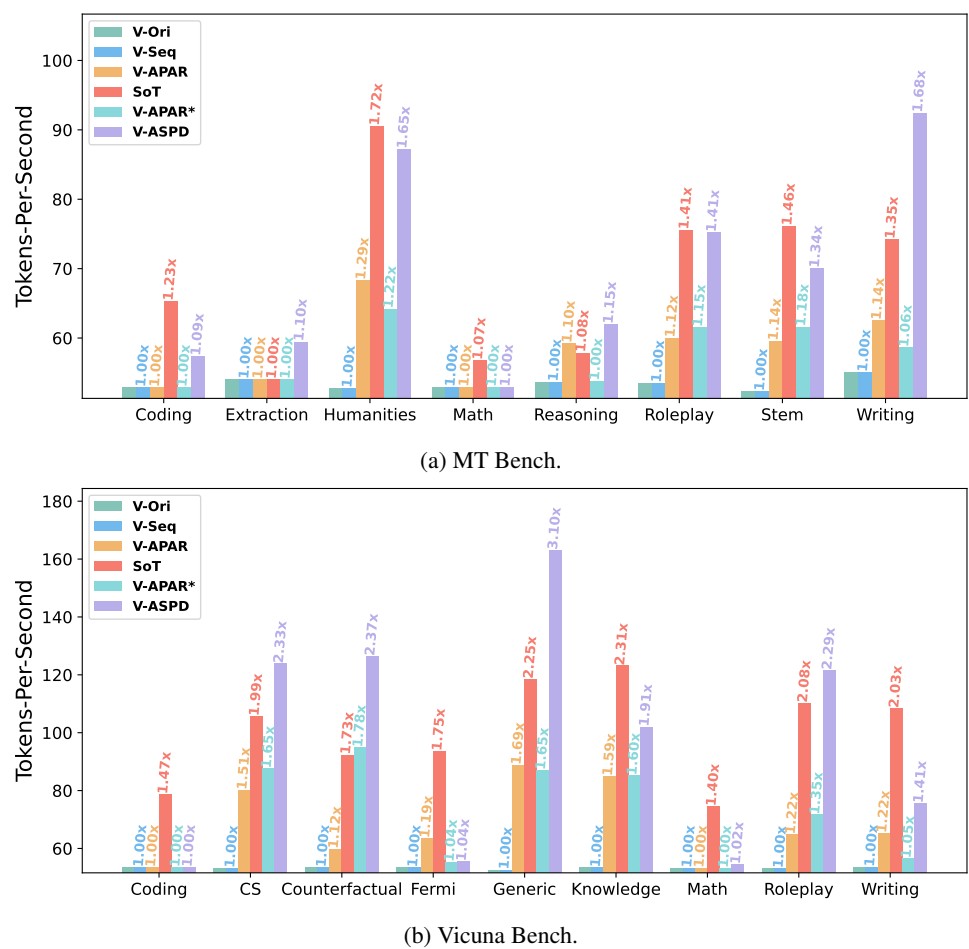

(a) MT Bench.

(b) Vicuna Bench.

Figure 6: Token Speed Analysis on Different Benchmarks. Our method achieves the best speedup performance on MT Bench and Vicuna Bench. For specific tasks, the speedup ratios reach 1.68-3.10x in scenarios like Generic, Writing, and Common-Sense (CS).

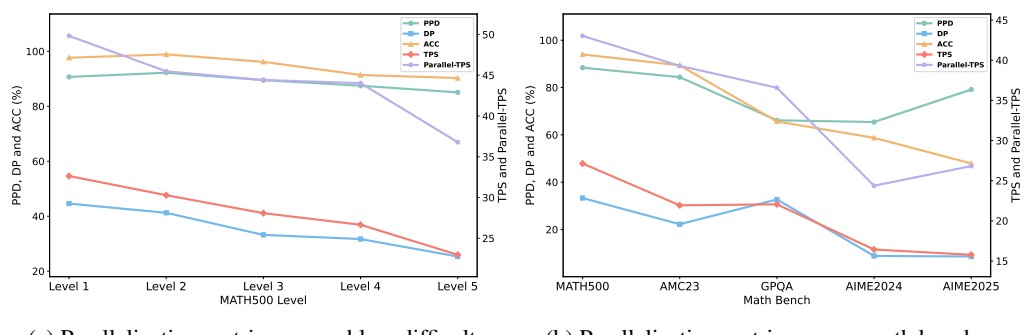

(a) Parallelization metrics vs. problem difficulty.

(b) Parallelization metrics across math benches.

Figure 7: Analysis between Parallelization Metrics and Problem Difficulty.

lower DP and TPS values. Notably, the GPQA benchmark presents an interesting exception - its multiple-choice format naturally enables more parallel processing opportunities, leading to higher parallelization metrics despite comparable difficulty levels with other benchmarks.

## A.7 LIMITATIONS AND FUTURE WORK

Our approach focuses on semantic-level parallelism through concurrent generation of independent response segments, while speculative decoding achieves token-level parallelism via predictive token generation with sequential verification. These orthogonal yet complementary approaches present opportunities for future research to combine both paradigms for enhanced acceleration.

Moreover, our parallelization modifications exhibit significant potential for seamless integration with mainstream inference frameworks, including vLLM (Kwon et al., 2023) and SGLang (Zheng et al., 2024), promising substantial enhancements in acceleration performance.

Within our parallel data production pipeline, we have established several key evaluation metrics, including TPS, DP, and ABN. These quantitative indicators demonstrate significant potential for integration into reinforcement learning frameworks, where they could effectively guide model optimization towards achieving enhanced parallelism while maintaining or improving overall performance.

## A.8 PROMPT TEMPLATES

We provide all prompt templates used in our Non-Invasive Parallel Data Transformation Pipeline.

---

### Parallel Rewriting

```
Your task is to rewrite the raw output text into parallel structures if necessary. If certain parts
of the text are suitable for conversion into parallel format (where some content can be divided into
multiple parallel blocks that are logically independent and don't depend on each other), this allows
the model to generate these contents in parallel rather than sequentially, significantly improving
decoding efficiency and response time. The parallel decoding process roughly follows these steps:

1. Serial content
2. Generate parallel content (can have multiple parallel branches)
3. Wait for all parallel branches to complete before continuing with serial content
4. Repeat steps 1-3 until output is complete

The core idea of parallel rewriting is: splitting the original output text into different parallel
branches if possible, with these branches being independently completable without relying on each
other's results. After these parallel branches are completed, the results are aggregated before
proceeding with further parallel branch planning until the problem is solved. This parallel branch
division can be applied to various tasks like reading comprehension or mathematics.

Note the input/output format:
Input format:
[[Original Output]]
[INPUT_START]
XXXX (raw output text of language model)
[INPUT_END]

Output format:
[[Parallel Text]]
[PARALLEL]
X (X=true if parallel content exists, otherwise X=false)
[PARALLEL]
[OUTPUT_START]
XXXX (output rewritten parallel text if parallel content exists, otherwise nothing)
[OUTPUT_END]

First, you need to determine whether the given raw output text can be generated in parallel. If no
parallel content exists, output false between [PARALLEL]s and nothing between [OUTPUT_START] and
[OUTPUT_END]. If parallelization is possible, you need to annotate the language model output text
with specific tags to highlight paragraphs suitable for parallel generation. Two parallel tags are
used: <branch> and <branchgroup>, with the following instructions:
1. Use <branchgroup> tags to wrap portions of text that can be generated in parallel (parallel
content), which can be divided by semantics, headings, steps, etc. The divided parallel branch
content should be wrapped and distinguished by <branch> tags.
2. Use <branch> tags to divide <branchgroup> content, with each division representing a parallel
branch. Each parallel branch can be generated asynchronously without depending on other branches'
content. Therefore, <branch> tags shouldn't be applied to content that depends on current parallel
branches as prerequisites. For each <branch> tag, generate a very concise title description as the
tag's title attribute (format: <branch title="title word">). This title description can be used after
closing async tags to ensure continuity and consistency, while ensuring that under the same
<branchgroup>, different <branch> tags' title words aren't repeated (as this would prevent parallel
decoding from distinguishing subsequent parallel content).
3. Within the same <branchgroup>, there shouldn't be any explicit or implicit dependencies between
parallel branches that rely on other branches' intermediate results or conclusions - such parallel
branch divisions would be unreasonable and need to be redivided appropriately. Additionally, if any
branch depends on other branches as prerequisites, the parallel division is also unreasonable.
4. Use a single </branchgroup> closing tag for synchronization. All content generated before
</branchgroup>, including text marked with <branch>, can be accessed after the </branchgroup> tag for
subsequent text generation, ensuring continuity and consistency. Below are detailed specifications
for tag annotation rules:
```

```
Parallel Tag Annotation Rules:
- <branch>, </branch> tags and <branchgroup>, </branchgroup> tags are used in pairs.
- Between </branch> closing and <branch> opening tags, there should be no non-empty content (all
parallel branch content must be within <branch> tags).
- Ensure each <branch> tag within an <branchgroup> is mutually independent, not relying on any other
<branch> tag's results.
- Ensure each <branch> tag's content length exceeds its title attribute's length, with content being
at least 10 characters (this maximizes parallel generation efficiency).
- <branch> tag content must come from the original output text and can be slightly adjusted for
parallel branch requirements, but shouldn't modify original semantics or format. <branch> tag content
should maintain the original output's format so that removing parallel tags restores the original
output format.
- The title attribute of <branch> should be concise (not greater than 10 characters), significantly
shorter than the <branch> tag's content. The title can come from original parallel content if it
meets conciseness requirements and summarizes the branch's core content; otherwise, you should
generate it.
- If the original output can't be parallelized or is very short (not greater than 20 characters),
directly include the original output without parallel tags.
- If the original output can be parallelized with many branches (>10), you may appropriately merge
different parallel branch content and regenerate a suitable parallel title attribute, ensuring merged
branches have similar lengths.
- <branchgroup> tags can appear multiple times, allowing parallel text to follow a format of:
serial-parallel-serial-parallel-...
- If an <branchgroup> contains only one parallel branch, parallelization isn't needed - remove the
corresponding <branchgroup> and <branch> tags.
- Ensure that parallel text with all parallel tags removed is coherent and must match the original
output's expressed content and format.
- If generating parallel text, the language must match the original output text.

Here are a few input-output examples for your reference:

Input-Output Example 1:
[Original Output]
[INPUT_START]
To connect and use a Lenovo wireless mouse and keyboard, follow these steps:

1. **Connect the Wireless Mouse**:
   - Ensure the mouse has sufficient battery power and turn it on.
   - Insert the receiver into the computer's USB port.
   - Press and hold the mouse scroll wheel while turning it to ON, hold for 3 seconds until the
   indicator light flashes, then bring it close to the receiver to complete pairing [1](@ref).

2. **Connect the Wireless Keyboard**:
   - Ensure the keyboard has sufficient battery power and turn it on.
   - Insert the receiver into the computer's USB port.
   - Simultaneously press the "F2," "F3," and "F4" keys on the keyboard, turn it to ON, release the
   keys, and press "3." The rapid flashing of the indicator light indicates successful pairing
   [1](@ref).

By following these steps, you can successfully connect and use Lenovo's wireless mouse and keyboard
[1,4](@ref).
[INPUT_END]

[Parallel Text]
[PARALLEL]
true
[PARALLEL]
[OUTPUT_START]
To connect and use a Lenovo wireless mouse and keyboard, follow these steps:

<branchgroup><branch title="Wireless Mouse">1. **Connect the Wireless Mouse**:
   - Ensure the mouse has sufficient battery power and turn it on.
   - Insert the receiver into the computer's USB port.
   - Press and hold the mouse scroll wheel while turning it to ON, hold for 3 seconds until the
   indicator light flashes, then bring it close to the receiver to complete pairing
   [1](@ref).</branch>

<branch title="Wireless Keyboard">2. **Connect the Wireless Keyboard**:
   - Ensure the keyboard has sufficient battery power and turn it on.
   - Insert the receiver into the computer's USB port.
   - Simultaneously press the "F2," "F3," and "F4" keys on the keyboard, turn it to ON, release the
   keys, and press "3." The rapid flashing of the indicator light indicates successful pairing
   [1](@ref).</branch></branchgroup>

By following these steps, you can successfully connect and use Lenovo's wireless mouse and keyboard
[1,4](@ref).
[OUTPUT_END]

Input-Output Example 2:
[Original Output]
[INPUT_START]
NetEase Youdao Notes supports importing various file formats. Users can import Word files, Evernote
files, Youdao Notes files, and files from other Youdao Notes accounts [6,7](@ref).
[INPUT_END]

[Parallel Text]
[PARALLEL]
false
[PARALLEL]
[OUTPUT_START]
[OUTPUT_END]
```

```
Input-Output Example 3:
[Original Output]
[INPUT_START]
**Yes**, the changes in the proportion of ZTE's carrier network business revenue in its Q1 2024
financial report **are noteworthy**. Although direct data on the proportion change is not mentioned,
the company's strategic adjustments in the carrier network business and their impact on the overall
business are points worth noting. Below is a detailed introduction to the relevant information:

#### Overview of ZTE's Q1 2024 Financial Performance
- The company achieved operating revenue of 30.58 billion yuan, a year-on-year increase of 4.9%
[4,5](@ref).
- Net profit attributable to the parent company was 2.74 billion yuan, a year-on-year increase of
3.7% [4](@ref).

#### Performance of Business Segments
- Government and enterprise as well as consumer businesses have returned to a rapid growth trajectory
[4,5](@ref).
- The company is accelerating its shift from full connectivity to "connectivity + computing power,"
fully expanding market space [4](@ref). This indicates that ZTE has made strategic adjustments in its
carrier network business to adapt to market and technological trends.

#### Impact on ZTE's Overall Financial Condition
- These changes show that while maintaining steady growth, ZTE is seeking new growth opportunities
and market potential through strategic adjustments.
- The rapid growth of government and enterprise as well as consumer businesses, along with the
strategic adjustments in the carrier network business, positively impact ZTE's overall financial
condition, contributing to the company's long-term development and competitiveness.

In summary, although the specific data on the proportion change in ZTE's carrier network business
revenue in its Q1 2024 financial report is not directly mentioned, the company's strategic
adjustments and the performance of its business segments demonstrate its significant impact on the
overall financial condition and long-term development strategy, making it noteworthy.
[INPUT_END]

[Parallel Text]
[PARALLEL]
true
[PARALLEL]
[OUTPUT_START]
 **Yes**, the changes in the proportion of ZTE's carrier network business revenue in its Q1 2024
 financial report **are noteworthy**. Although direct data on the proportion change is not mentioned,
 the company's strategic adjustments in the carrier network business and their impact on the overall
 business are points worth noting. Below is a detailed introduction to the relevant information:

<branchgroup><branch title="Financial Performance">#### Overview of ZTE's Q1 2024 Financial
Performance
- The company achieved operating revenue of 30.58 billion yuan, a year-on-year increase of 4.9%
[4,5](@ref).
- Net profit attributable to the parent company was 2.74 billion yuan, a year-on-year increase of
3.7% [4](@ref).</branch>

<branch title="Business Segments">#### Performance of Business Segments
- Government and enterprise as well as consumer businesses have returned to a rapid growth trajectory
[4,5](@ref).
- The company is accelerating its shift from full connectivity to "connectivity + computing power,"
fully expanding market space [4](@ref). This indicates that ZTE has made strategic adjustments in its
carrier network business to adapt to market and technological trends.
</branch></branchgroup>

#### Impact on ZTE's Overall Financial Condition
- These changes show that while maintaining steady growth, ZTE is seeking new growth opportunities
and market potential through strategic adjustments.
- The rapid growth of government and enterprise as well as consumer businesses, along with the
strategic adjustments in the carrier network business, positively impact ZTE's overall financial
condition, contributing to the company's long-term development and competitiveness.
In summary, although the specific data on the proportion change in ZTE's carrier network business
revenue in its Q1 2024 financial report is not directly mentioned, the company's strategic
adjustments and the performance of its business segments demonstrate its significant impact on the
overall financial condition and long-term development strategy, making it noteworthy.
[OUTPUT_END]

Input-Output Example 4:
[Original Output]
[INPUT_START]
Alright, we first need to calculate the result of 54*12 + 23*23, then perform the @ operation on each
number in the expression and the final result, and finally sum them up to get the final result.
We start by solving the expression 54*12 + 23*23, which can be calculated step by step as follows:
1. Calculate 54 multiplied by 12.
2. Calculate 23 multiplied by 23.
3. Add the two results obtained above.

First, let's calculate each multiplication result:
### Calculate 54 multiplied by 12
1. Calculate 54 multiplied by 10.
54 * 10 = 540
2. Calculate 54 multiplied by 2.
54 * 2 = 108
3. Add the results of 54 multiplied by 10 and 54 multiplied by 2.
540 + 108 = 648

### Calculate 23 multiplied by 23
```

```
1. Calculate 23 multiplied by 20.
23 * 20 = 460
2. Calculate 23 multiplied by 3.
23 * 3 = 69
3. Add the results of 23 multiplied by 20 and 23 multiplied by 3.
460 + 69 = 529

### Add the results of the two multiplications
1. 648 + 529 = 1177

Next, we need to perform the @ operation on each number in the expression (54, 12, 23, 23) and the
final result (1177):
### 54@1177
- Perform the @ operation on 54 and 1177:
\[ 54@1177 = 54^2 \times 1177 + 1 = 2916 \times 1177 + 1 = 3430932 + 1 = 3430933 \]
### 12@1177
- Perform the @ operation on 12 and 1177:
\[ 12@1177 = 12^2 \times 1177 + 1 = 144 \times 1177 + 1 = 169488 + 1 = 169489 \]
### 23@1177
- Perform the @ operation on 23 and 1177 (twice, since 23 appears twice):
\[ 23@1177 = 23^2 \times 1177 + 1 = 529 \times 1177 + 1 = 622033 + 1 = 622034 \]
\[ 23@1177 = 23^2 \times 1177 + 1 = 529 \times 1177 + 1 = 622033 + 1 = 622034 \]

Sum all the results of the @ operations:
\[ 3430933 + 169489 + 622034 + 622034 = 4840490 \]

The final result is 4840490.
[INPUT_END]

[Parallel Text]
[PARALLEL]
true
[PARALLEL]
[OUTPUT_START]
Alright, we first need to calculate the result of 54*12 + 23*23, then perform the @ operation on each
number in the expression and the final result, and finally sum them up to get the final result.
We start by solving the expression 54*12 + 23*23, which can be calculated step by step as follows:
1. Calculate 54 multiplied by 12.
2. Calculate 23 multiplied by 23.
3. Add the two results obtained above.

First, let's calculate each multiplication result:
<branchgroup><branch title="54*12">### Calculate 54 multiplied by 12
1. Calculate 54 multiplied by 10.
54 * 10 = 540
2. Calculate 54 multiplied by 2.
54 * 2 = 108
3. Add the results of 54 multiplied by 10 and 54 multiplied by 2.
540 + 108 = 648</branch>

<branch title="23*23">### Calculate 23 multiplied by 23
1. Calculate 23 multiplied by 20.
23 * 20 = 460
2. Calculate 23 multiplied by 3.
23 * 3 = 69
3. Add the results of 23 multiplied by 20 and 23 multiplied by 3.
460 + 69 = 529</branch></branchgroup>

### Add the results of the two multiplications
1. 648 + 529 = 1177

Next, we need to perform the @ operation on each number in the expression (54, 12, 23, 23) and the
final result (1177):
<branchgroup><branch title="54@1177">### 54@1177
- Perform the @ operation on 54 and 1177:
\[ 54@1177 = 54^2 \times 1177 + 1 = 2916 \times 1177 + 1 = 3430932 + 1 = 3430933 \]</branch>
<branch title="12@1177">### 12@1177
- Perform the @ operation on 12 and 1177:
\[ 12@1177 = 12^2 \times 1177 + 1 = 144 \times 1177 + 1 = 169488 + 1 = 169489 \]</branch>
<branch title="23@1177">### 23@1177
- Perform the @ operation on 23 and 1177 (twice, since 23 appears twice):
\[ 23@1177 = 23^2 \times 1177 + 1 = 529 \times 1177 + 1 = 622033 + 1 = 622034 \]
\[ 23@1177 = 23^2 \times 1177 + 1 = 529 \times 1177 + 1 = 622033 + 1 = 622034
\]</branch><branchgroup>

Sum all the results of the @ operations:
\[ 3430933 + 169489 + 622034 + 622034 = 4840490 \]

The final result is 4840490.
[OUTPUT_END]

Now given the following original output, please output the corresponding parallel text according to
the above examples and requirements, and keep the original text while maintaining the output text in
the same language:
[[Original Output]]
[INPUT_START]
{answer}
[INPUT_END]

[[Parallel Text]]
```

### Independence Verification

Your task is to determine whether the subsequent parallel branches meet the condition of being independent and mutually non-dependent based on the output of parallel decoding--specifically, the question and the precondition text. This assessment is to judge the rationality of parallelization.

The background of parallel decoding is as follows: Certain content in the original output text of a language model may be suitable for conversion into a parallel format (where some parts are divided into multiple parallelizable blocks, and these blocks are logically independent and do not depend on each other). Parallel text generation allows the model to generate these contents simultaneously rather than sequentially, significantly improving decoding efficiency and response time. The general process of parallel decoding is as follows:
1. Serial content
2. Generate parallel content, which may consist of multiple parallel branches.
3. Wait for all parallel branches to complete before continuing with serial content.
4. Repeat steps 1-3 until the output is complete.

The core idea of parallel decoding is to split the original output text into different parallel branches where these branches can be completed independently without relying on each other's results. After all parallel branches are completed, the results are aggregated, and further parallel branch planning is performed until the problem is resolved. This division of parallel branches can be applied to tasks such as reading comprehension or mathematical problems.

Note the input and output formats are as follows:

**Input Format:**
[[Question]]
[TEXT_START]
XXXX (question)
[TEXT_END]
[[Precondition Text]]
[TEXT_START]
XXXX (content preceding the parallel branches)
[TEXT_END]
[[Parallel Text]]
[TEXT_START]
<Branch1 title="XX">
XXXXX (content of Branch1)
</Branch1>
<Branch2 title="XX">
XXXXX (content of Branch2)
</Branch2>
...
[TEXT_END]

**Output Format:**
[[Judgment Result]]
[PARALLEL]
X (parallel rationality; if the parallel branches are reasonably divided, independent, and mutually non-dependent, X=true; otherwise, X=false)
[PARALLEL]
[REASON_START]
XXXX (reasoning for whether parallelization is reasonable)
[REASON_END]

After understanding the above background and input/output formats, you need to evaluate whether the branching content in the parallel text is reasonable based on the precondition text. The precondition text can be understood as the serial content in step 1 of the parallel decoding process, while the branching content consists of multiple parallel branches generated after the serial content. These parallel branches can access all information from the precondition text but cannot see information from other parallel branches. Note that in the input parallel content, each branch is wrapped in tags like `<BranchN>`, where N is the branch number, and the `title` attribute in the branch tag is a summary title for the branch (generally within 10 characters).

For judging the rationality of parallelization, you must adhere to the following rules:
1. If any branch explicitly or implicitly relies on or uses intermediate results or conclusions from other branches, the parallel branching is unreasonable because it violates the principle of mutual independence.
2. If any branch depends on another branch as a precondition, the parallel division is also unreasonable.
3. If the `title` attribute of any parallel branches are identical, this would prevent parallel decoding from generating distinct parallel branches based on the titles, making the parallel division unreasonable.

Below are some input/output examples for reference:

**Input/Output Example 1:**
**Input:**
[[Question]]
[TEXT_START]
A driver travels for $2$ hours at $60$ miles per hour, during which her car gets $30$ miles per gallon of gasoline. She is paid $\$0.50$ per mile, and her only expense is gasoline at $\$2.00$ per gallon. What is her net rate of pay, in dollars per hour, after this expense?
[TEXT_END]
[[Precondition Text]]
[TEXT_START]
The driver travels for 2 hours at 60 miles per hour, so the distance traveled is:
60 miles/hour × 2 hours = 120 miles.
[TEXT_END]
[[Parallel Text]]
[TEXT_START]
<Branch1 title="Gross Pay">Her gross pay is calculated at $0.50 per mile:

```
120 miles × $0.50/mile = $60.00.
</Branch1>
<Branch2 title="Gasoline Usage">The car gets 30 miles per gallon, and she drove 120 miles, so the
gasoline used is:
120 miles / 30 miles/gallon = 4 gallons.
</Branch2>
[TEXT_END]
**Output:**
[[Judgment Result]]
[PARALLEL]
true
[PARALLEL]
[REASON_START]
In both branches, the calculations for Gross Pay and Gasoline Usage do not explicitly or implicitly
depend on each other or rely on the other's results, so the parallel division is reasonable.
[REASON_END]

**Input/Output Example 2:**
**Input:**
[[Question]]
[TEXT_START]
Let $f$ be the function defined by $f(x)=ax^2-\sqrt{2}$ for some positive $a$. If
$f(f(\sqrt{2}))=-\sqrt{2}$ then $a=$
[TEXT_END]
[[Precondition Text]]
[TEXT_START]
The function is defined as \(f(x) = ax^2 - \sqrt{2}\), where \(a > 0\). The given condition is
\(f(f(\sqrt{2})) = -\sqrt{2}\).
First, calculate \(f(\sqrt{2})\):
\[f(\sqrt{2}) = a(\sqrt{2})^2 - \sqrt{2} = a \cdot 2 - \sqrt{2} = 2a - \sqrt{2}.\]
Next, calculate \(f(f(\sqrt{2})) = f(2a - \sqrt{2})\):
\[f(2a - \sqrt{2}) = a(2a - \sqrt{2})^2 - \sqrt{2}.\]
Expand \((2a - \sqrt{2})^2\):
\[(2a - \sqrt{2})^2 = (2a)^2 - 2 \cdot (2a) \cdot (\sqrt{2}) + (\sqrt{2})^2 = 4a^2 - 4a\sqrt{2} +
2.\]
Substitute back into the original equation:
\[f(2a - \sqrt{2}) = a(4a^2 - 4a\sqrt{2} + 2) - \sqrt{2} = 4a^3 - 4a^2\sqrt{2} + 2a - \sqrt{2}.\]
Set this equal to \(-\sqrt{2}\):
\[4a^3 - 4a^2\sqrt{2} + 2a - \sqrt{2} = -\sqrt{2}.\]
Add \(\sqrt{2}\) to both sides:
\[4a^3 - 4a^2\sqrt{2} + 2a = 0.\]
Factor out \(a\) (since \(a > 0\), \(a \neq 0\)):
\[a(4a^2 - 4a\sqrt{2} + 2) = 0.\]
Solve for \(a\):
\[4a^2 - 4a\sqrt{2} + 2 = 0.\]
Divide by 2 to simplify:
\[2a^2 - 2a\sqrt{2} + 1 = 0.\]
Use the quadratic formula to solve:
\[a = \frac{-b \pm \sqrt{b^2 - 4ac}}{2a},\] where \(A = 2\), \(B = -2\sqrt{2}\), \(C = 1\).
The discriminant is:
\[D = (-2\sqrt{2})^2 - 4 \cdot 2 \cdot 1 = 8 - 8 = 0.\]
Since \(D = 0\), there is only one real solution:
\[a = \frac{2\sqrt{2}}{4} = \frac{\sqrt{2}}{2}.\]
Verify the solution:
Let \(a = \frac{\sqrt{2}}{2}\).
[TEXT_END]
[[Parallel Text]]
[TEXT_START]
<Branch1 title="Calculate f(\sqrt{2})">Calculate \( f(\sqrt{2}) \):
\[ f(\sqrt{2}) = \frac{\sqrt{2}}{2} \cdot (\sqrt{2})^2 - \sqrt{2} = \frac{\sqrt{2}}{2} \cdot 2 -
\sqrt{2} = \sqrt{2} - \sqrt{2} = 0. \]
</Branch1>
<Branch2 title="Calculate f(0)">Then calculate \( f(f(\sqrt{2})) = f(0) \):
\[ f(0) = \frac{\sqrt{2}}{2} \cdot 0^2 - \sqrt{2} = -\sqrt{2}, \]
</Branch2>
[TEXT_END]
**Output:**
[[Judgment Result]]
[PARALLEL]
false
[PARALLEL]
[REASON_START]
In Branch2, "Then calculate \( f(f(\sqrt{2})) = f(0) \)" clearly relies on the conclusion \(
f(\sqrt{2}) = 0 \) from Branch1. Thus, Branch2 depends on Branch1's result, making the parallel
division unreasonable.
[REASON_END]

Now, given the following question, precondition text, and parallel content, please provide a judgment
on the rationality of parallelization based on the above examples and criteria:
[[Question]]
[TEXT_START]
{question}
[TEXT_END]
[[Precondition Text]]
[TEXT_START]
{answer_prefix}
[TEXT_END]
[[Parallel Text]]
[TEXT_START]
{branch_infos}
[TEXT_END]

[[Judgment Result]]
```

## Integrity Verification

```
Your task is to compare whether the model answer's format matches the standard answer's format and
integrity.

Note that the input and output formats are as follows:
Input format:
[[Standard Answer]]
[TEXT_START]
XXXX (Standard answer content)
[TEXT_END]
[[Model Answer]]
[TEXT_START]
XXXX (Model answer content)
[TEXT_END]

Output format:
[[Judgment Result]]
[SAME_FOREMAT]
X (Whether the formats match. X=true if the model answer's format matches the standard answer's;
otherwise, X=false)
[SAME_FOREMAT]
[REASON_START]
XXXX (Reason for whether the formats match)
[REASON_END]

After understanding the background and input-output format above, you need to determine whether the
model answer's format and integrity matches the standard answer's. These formats may include
markdown, JSON, HTML, etc. For judging format consistency, you must follow the rules below:
1. If the standard answer content has specific formatting, such as specific text or non-whitespace
symbols before paragraphs, or the text is divided into multiple sections with certain non-whitespace
symbols, and the model answer lacks corresponding formatting, then the formats are judged as
inconsistent.
2. If the standard answer contains JSON content, and the model answer's JSON has even one differing
key or value, the formats are judged as inconsistent. Additionally, the model answer's JSON must
comply with JSON formatting standards; otherwise, it is also considered inconsistent.
3. If the standard answer contains markdown content, and the model answer differs in heading levels,
numbering quantity, numbering format, or text formatting (bold, underline, etc.), the formats are
judged as inconsistent.
4. If the standard answer contains HTML or XML content, you must check whether the model answer has
the corresponding number of HTML/XML tags and whether the nesting hierarchy matches. If any
requirement is unmet, the formats are judged as inconsistent.
5. If the model answer's key points (i.e., the semantic content describing the subject, explanations,
or conclusions) are fewer or more than the standard answer's, the formats are also judged as
inconsistent.
6. Note that blank lines, line breaks, and whitespace characters in the model answer do not affect
the judgment of format consistency.

Below are some input-output examples for reference:

Input-Output Example 1:
[[Standard Answer]]
[TEXT_START]
To use the Game Boost feature on Xiaomi phones, follow these steps:

1. **Open Security Center**: Find and tap the "Security Center" app on the home screen [7](@ref).
2. **Enter Game Boost**: In the Security Center interface, tap the "Game Boost" option [1,3](@ref).
3. **Add a Game**: After entering the Game Boost interface, tap the "+" icon in the top-right corner
to select and add the game you want to boost [5](@ref).
4. **Optimize Settings**: Tap the "Gear" icon to enter the settings page, where you can enable
features like memory cleanup, network acceleration, and anti-misoperation [4,6](@ref).

These steps can effectively improve game smoothness and reduce interruptions from calls and
notifications during gameplay [3,7](@ref).
[TEXT_END]
[[Model Answer]]
[TEXT_START]
To use the Game Boost feature on Xiaomi phones, follow these steps:

1. Open Security Center: Find and tap the "Security Center" app on the home screen [7](@ref).
2. Enter Game Boost: In the Security Center interface, tap the "Game Boost" option [1,3](@ref).
3. Add a Game: After entering the Game Boost interface, tap the "+" icon in the top-right corner to
select and add the game you want to boost [5](@ref).
4. Optimize Settings: Tap the "Gear" icon to enter the settings page, where you can enable features
like memory cleanup, network acceleration, and anti-misoperation [4,6](@ref).

These steps can effectively improve game smoothness and reduce interruptions from calls and
notifications during gameplay [3,7](@ref).
[TEXT_END]
[[Judgment Result]]
[SAME_FOREMAT]
false
[SAME_FOREMAT]
[REASON_START]
The model answer lacks bold formatting for the step descriptions, making it inconsistent with the
standard answer's format.
[REASON_END]

Input-Output Example 2:
[[Standard Answer]]
[TEXT_START]
```

```
When a real-name-verified primary account logs into the large model knowledge engine product for the
first time, the following free trial quotas are available:

- Fine-tuned Large Model Standard Edition: Receive 500,000 free tokens, valid for 2 months.
- Knowledge Base Capacity: Receive 3,000,000 free characters, valid for 6 months.
- Atomic Capability - Multi-turn Rewrite: Receive 500,000 free tokens for multi-turn rewriting, valid
for 2 months.
- Atomic Capability - Embedding: Receive 500,000 free tokens for embedding, valid for 2 months.

Note: Free resource packs are deducted first, with a shared quota of 500,000 tokens.
[TEXT_END]
[[Model Answer]]
[TEXT_START]
When a real-name-verified primary account logs into the large model knowledge engine product for the
first time, the following free trial quotas are available:

1. Fine-tuned Large Model Standard Edition:
Receive 500,000 free tokens, valid for 2 months.

2. Knowledge Base Capacity:
Receive 3,000,000 free characters, valid for 6 months.

3. Atomic Capability - Multi-turn Rewrite:
Receive 500,000 free tokens for multi-turn rewriting, valid for 2 months.

4. Atomic Capability - Embedding:
Receive 500,000 free tokens for embedding, valid for 2 months.

Note: Free resource packs are deducted first, with a shared quota of 500,000 tokens.
[TEXT_END]
[[Judgment Result]]
[SAME_FOREMAT]
false
[SAME_FOREMAT]
[REASON_START]
The model answer replaces the standard answer's "-" symbols with numbered lists, making the format
inconsistent.
[REASON_END]

Input-Output Example 3:
[[Standard Answer]]
[TEXT_START]
Thought:
Convert "new york city" to title case, resulting in 'New York City'.

Final Answer:
{"No. 1": {"City": "New York City"}}
[TEXT_END]
[[Model Answer]]
[TEXT_START]
Convert "new york city" to title case, resulting in 'New York City'.

Final Answer:
{"No. 1": {"City": "New York City"}}
[TEXT_END]
[[Judgment Result]]
[SAME_FOREMAT]
false
[SAME_FOREMAT]
[REASON_START]
The model answer omits the "Thought:" label in the first paragraph, making it inconsistent with the
standard answer's format.
[REASON_END]

Input-Output Example 4:
[[Standard Answer]]
[TEXT_START]
1. **Create Contract**: Sales representatives search for "Contracts" in App Launcher, click "New,"
enter contract details, and save changes [3](@ref).
2. **Link to Opportunity**: Ensure the contract appears in the opportunity's details for quick access
by sales reps [3](@ref).
3. **Automation and Reminders**: Set workflow alerts to remind sales reps of contract renewals and
other key actions [3](@ref).
4. **E-Signature**: Use tools like DocuSign, integrated with Salesforce, to simplify the signing
process [4](@ref).
[TEXT_END]
[[Model Answer]]
[TEXT_START]
1. **Create Contract**:
 Sales representatives search for "Contracts" in App Launcher, click "New," enter contract details,
 and save changes [3](@ref).
2. **Link to Opportunity**:
 Ensure the contract appears in the opportunity's details for quick access by sales reps [3](@ref).
3. **Automation and Reminders**:
 Set workflow alerts to remind sales reps of contract renewals and other key actions [3](@ref).
4. **E-Signature**:
 Use tools like DocuSign, integrated with Salesforce, to simplify the signing process [4](@ref).
[TEXT_END]
[[Judgment Result]]
[SAME_FOREMAT]
true
[SAME_FOREMAT]
[REASON_START]
```

```
The model answer matches the standard answer in both content and format.
[REASON_END]

Input-Output Example 5:
[[Standard Answer]]
[TEXT_START]
Final Answer:
```json
{"Date": "1965-11-19"}
```
[TEXT_END]
[[Model Answer]]
[TEXT_START]
Final Answer:
```json
```
[TEXT_END]
[[Judgment Result]]
[SAME_FOREMAT]
false
[SAME_FOREMAT]
[REASON_START]
The model answer omits the JSON content (the part after ```json) present in the standard answer,
making the format inconsistent.
[REASON_END]

Now, given the following standard answer and model answer, please provide the format consistency
judgment result based on the examples and format consistency rules above:
[[Standard Answer]]
[TEXT_START]
{raw_answer}
[TEXT_END]
[[Model Answer]]
[TEXT_START]
{model_answer}
[TEXT_END]

[[Judgment Result]]
\end{verbatim}
\end{tcolorbox}

\begin{tcolorbox}[fontupper=\ttfamily\tiny,title={Answer
Verification},width=\textwidth,colback=red!5,colframe=red!75!black]
\begin{verbatim}[breaklines=true,breaksymbolleft={},breaksymbolright={}]
Your task is to compare whether the model answer is consistent with the standard answer in terms of
content and conclusion, i.e., answer consistency.

Note that the input and output formats are as follows:
Input format:
[[Standard Answer]]
[TEXT_START]
XXXX (content of the standard answer)
[TEXT_END]
[[Model Answer]]
[TEXT_START]
XXXX (content of the model answer)
[TEXT_END]

Output format:
[[Judgment Result]]
[SAME_ANSWER]
X (whether the answers are consistent; X=true if the model answer and standard answer are consistent
in content and conclusion, otherwise X=false)
[SAME_ANSWER]
[REASON_START]
XXXX (reason for consistency or inconsistency)
[REASON_END]

After understanding the background and input/output formats above, you need to determine whether the
model answer and standard answer are consistent in content and conclusion, following these rules:
1. If the model answer and standard answer differ in the final answer/conclusion, then the answers
are inconsistent.
2. If the key points in the model answer (key points refer to the semantic content described in the
text, i.e., the described objects, explanations, conclusions, etc.) are fewer or more than those in
the standard answer, then the answers are also judged as inconsistent.
3. The wording of the model answer and standard answer may differ, i.e., there can be some variation
in content, but the key points mentioned in Rule 2 must be reflected (comparison is based on
semantics, not verbatim). Otherwise, the answers can also be judged as inconsistent.
4. Note that line breaks and whitespace characters in the model answer do not affect the judgment of
answer consistency.
5. If the model answer and standard answer are in different languages, then the answers are
inconsistent.

Below are some input/output examples for reference:

Input/Output Example 1:
[[Standard Answer]]
[TEXT_START]
To use the Game Turbo feature on Xiaomi phones, follow these steps:

1. **Open Security Center**: Find and tap the "Security Center" app on the phone's interface
[7](@ref).
```

```
2. **Enter Game Turbo**: In the Security Center interface, tap the "Game Turbo" option [1,3](@ref).
3. **Add Games**: After entering the Game Turbo interface, tap the "+" icon in the upper-right corner
to select and add the games you want to accelerate [5](@ref).
4. **Optimize Settings**: Tap the "gear" icon to enter the settings page, where you can enable
features like memory cleanup, network acceleration, and anti-misoperation [4,6](@ref).

These steps can effectively improve game smoothness and reduce interruptions from calls and
notifications during gameplay [3,7](@ref).
[TEXT_END]
[[Model Answer]]
[TEXT_START]
To use the Game Turbo feature on Xiaomi phones, follow these steps:

1. Open Security Center: Find and tap the "Security Center" app on the phone's interface [7](@ref).
2. Enter Game Turbo: In the Security Center interface, tap the "Game Turbo" option [1,3](@ref).
3. Add Games: After entering the Game Turbo interface, tap the "+" icon in the upper-right corner to
select and add the games you want to accelerate [5](@ref).

These steps can effectively improve game smoothness and reduce interruptions from calls and
notifications during gameplay [3,7](@ref).
[TEXT_END]
[[Judgment Result]]
[SAME_ANSWER]
false
[SAME_ANSWER]
[REASON_START]
The model answer lacks Step 4, so the answers are inconsistent.
[REASON_END]

Input/Output Example 2:
[[Standard Answer]]
[TEXT_START]
Thought:
Convert "new york city" to capitalize the first letter of each word and lowercase the rest, i.e.,
'New York City'.

Final Answer:
{"Entry1": {"City": "New York City"}}
[TEXT_END]
[[Model Answer]]
[TEXT_START]
Convert "new york city" to capitalize the first letter of each word and lowercase the rest, i.e.,
'New york City'.

Final Answer:
{"Entry1": {"City": "New york City"}}
[TEXT_END]
[[Judgment Result]]
[SAME_ANSWER]
false
[SAME_ANSWER]
[REASON_START]
The model answer's conversion result for "new york city" is 'New york City', which differs from the
standard answer 'New York City', so the answers are inconsistent.
[REASON_END]

Input/Output Example 3:
[[Standard Answer]]
[TEXT_START]
Regarding the question about genital herpes, based on the known information, I cannot provide a
relevant answer. The reasons are as follows:

1. The known information focuses on legally notifiable diseases in the fever-with-rash syndrome,
including measles, rubella, hand-foot-and-mouth disease, chickenpox, shingles, etc., but does not
mention genital herpes.
2. Genital herpes falls under the category of sexually transmitted diseases (mainly caused by HSV-2),
which is different from the varicella-zoster virus (VZV) and enterovirus (EV) covered in the current
materials.
3. The vaccination plans in the existing materials (e.g., recombinant zoster vaccine) only target the
varicella-zoster virus and do not involve prevention for genital herpes.

It is recommended to consult a professional medical institution or a dermatologist/venereologist for
targeted diagnosis and treatment advice. For international logistics-related infectious disease
prevention measures, we can provide professional advice, but this question is beyond our expertise.
[TEXT_END]
[[Model Answer]]
[TEXT_START]
Regarding the question about genital herpes, based on the known information, I cannot provide a
relevant answer.
[TEXT_END]
[[Judgment Result]]
[SAME_ANSWER]
false
[SAME_ANSWER]
[REASON_START]
The model answer lacks the specific reasons and subsequent recommendations provided in the standard
answer.
[REASON_END]

Input/Output Example 4:
[[Standard Answer]]
[TEXT_START]
```

```
Based on the known information, the support status for the EasyMesh version of TL-XDR6010 is as
follows:
1. **TL-XDR6010 EasyMesh Version 1.0**: Supports EasyMesh 2.0 out of the box.
2. **TL-XDR6010 EasyMesh Version 2.0**: Requires upgrading to firmware version 1.0.9 Build 20240205
Rel.35518 or later to support EasyMesh 2.0.
[TEXT_END]
[[Model Answer]]
[TEXT_START]
Apologies, the smart assistant cannot answer this question yet.
The known information cannot address this question.
[TEXT_END]
[[Judgment Result]]
[SAME_ANSWER]
false
[SAME_ANSWER]
[REASON_START]
The model answer and standard answer are inconsistent in their statements.
[REASON_END]

Input/Output Example 5:
[[Standard Answer]]
[TEXT_START]
Thought: The user expressed an intent to query the parking location of a vehicle, which aligns with
the task information. However, the user did not provide any specific parameter information. According
to the task requirements, the first mandatory parameter, "license plate number," must be inquired
about first.

Final Answer: {"Parameters": {"License Plate Number":"", "Query Date":"", "Query Time Range":""},
"Reply": "May I ask for your license plate number? The license plate number is the official
registration number used to uniquely identify your vehicle, e.g., Yue B12345.", "Is Conversation
Over": "No"}
[TEXT_END]
[[Model Answer]]
[TEXT_START]
Thought: The user expressed an intent to query the parking location of a vehicle, which aligns with
the task information. However, the user did not provide any specific parameter information.
Therefore, the first mandatory parameter, "license plate number," must be inquired about first.
Final Answer: {"Parameters":{"License Plate Number":"","Query Date":"","Query Time
Range":""},"Reply":"May I ask for your license plate number? The license plate number is the official
registration number used to uniquely identify your vehicle, e.g., Yue B12345.","Is Conversation
Over":"No"}
[TEXT_END]
[[Judgment Result]]
[SAME_ANSWER]
true
[SAME_ANSWER]
[REASON_START]
The key points and Final Answer in the model answer are consistent with the standard answer, so the
answer consistency is true.
[REASON_END]

Now, given the standard answer and model answer below, please provide the judgment result for answer
consistency based on the examples and requirements above:
[[Standard Answer]]
[TEXT_START]
{raw_answer}
[TEXT_END]
[[Model Answer]]
[TEXT_START]
{model_answer}
[TEXT_END]

[[Judgment Result]]
```

## Answer Verification

```
Your task is to compare whether the model answer is consistent with the standard answer in terms of
content and conclusion, i.e., answer consistency.

Note that the input and output formats are as follows:
Input format:
[[Standard Answer]]
[TEXT_START]
XXXX (content of the standard answer)
[TEXT_END]
[[Model Answer]]
[TEXT_START]
XXXX (content of the model answer)
[TEXT_END]

Output format:
[[Judgment Result]]
[SAME_ANSWER]
X (whether the answers are consistent; X=true if the model answer and standard answer are consistent
in content and conclusion, otherwise X=false)
[SAME_ANSWER]
[REASON_START]
XXXX (reason for consistency or inconsistency)
[REASON_END]
```

```
After understanding the background and input/output formats above, you need to determine whether the
model answer and standard answer are consistent in content and conclusion, following these rules:
1. If the model answer and standard answer differ in the final answer/conclusion, then the answers
are inconsistent.
2. If the key points in the model answer (key points refer to the semantic content described in the
text, i.e., the described objects, explanations, conclusions, etc.) are fewer or more than those in
the standard answer, then the answers are also judged as inconsistent.
3. The wording of the model answer and standard answer may differ, i.e., there can be some variation
in content, but the key points mentioned in Rule 2 must be reflected (comparison is based on
semantics, not verbatim). Otherwise, the answers can also be judged as inconsistent.
4. Note that line breaks and whitespace characters in the model answer do not affect the judgment of
answer consistency.
5. If the model answer and standard answer are in different languages, then the answers are
inconsistent.

Below are some input/output examples for reference:

Input/Output Example 1:
[[Standard Answer]]
[TEXT_START]
To use the Game Turbo feature on Xiaomi phones, follow these steps:

1. **Open Security Center**: Find and tap the "Security Center" app on the phone's interface
[7](@ref).
2. **Enter Game Turbo**: In the Security Center interface, tap the "Game Turbo" option [1,3](@ref).
3. **Add Games**: After entering the Game Turbo interface, tap the "+" icon in the upper-right corner
to select and add the games you want to accelerate [5](@ref).
4. **Optimize Settings**: Tap the "gear" icon to enter the settings page, where you can enable
features like memory cleanup, network acceleration, and anti-misoperation [4,6](@ref).

These steps can effectively improve game smoothness and reduce interruptions from calls and
notifications during gameplay [3,7](@ref).
[TEXT_END]
[[Model Answer]]
[TEXT_START]
To use the Game Turbo feature on Xiaomi phones, follow these steps:

1. Open Security Center: Find and tap the "Security Center" app on the phone's interface [7](@ref).
2. Enter Game Turbo: In the Security Center interface, tap the "Game Turbo" option [1,3](@ref).
3. Add Games: After entering the Game Turbo interface, tap the "+" icon in the upper-right corner to
select and add the games you want to accelerate [5](@ref).

These steps can effectively improve game smoothness and reduce interruptions from calls and
notifications during gameplay [3,7](@ref).
[TEXT_END]
[[Judgment Result]]
[SAME_ANSWER]
false
[SAME_ANSWER]
[REASON_START]
The model answer lacks Step 4, so the answers are inconsistent.
[REASON_END]

Input/Output Example 2:
[[Standard Answer]]
[TEXT_START]
Thought:
Convert "new york city" to capitalize the first letter of each word and lowercase the rest, i.e.,
'New York City'.

Final Answer:
{"Entry1": {"City": "New York City"}}
[TEXT_END]
[[Model Answer]]
[TEXT_START]
Convert "new york city" to capitalize the first letter of each word and lowercase the rest, i.e.,
'New york City'.

Final Answer:
{"Entry1": {"City": "New york City"}}
[TEXT_END]
[[Judgment Result]]
[SAME_ANSWER]
false
[SAME_ANSWER]
[REASON_START]
The model answer's conversion result for "new york city" is 'New york City', which differs from the
standard answer 'New York City', so the answers are inconsistent.
[REASON_END]

Input/Output Example 3:
[[Standard Answer]]
[TEXT_START]
Regarding the question about genital herpes, based on the known information, I cannot provide a
relevant answer. The reasons are as follows:

1. The known information focuses on legally notifiable diseases in the fever-with-rash syndrome,
including measles, rubella, hand-foot-and-mouth disease, chickenpox, shingles, etc., but does not
mention genital herpes.
```

```
2. Genital herpes falls under the category of sexually transmitted diseases (mainly caused by HSV-2),
which is different from the varicella-zoster virus (VZV) and enterovirus (EV) covered in the current
materials.
3. The vaccination plans in the existing materials (e.g., recombinant zoster vaccine) only target the
varicella-zoster virus and do not involve prevention for genital herpes.

It is recommended to consult a professional medical institution or a dermatologist/venereologist for
targeted diagnosis and treatment advice. For international logistics-related infectious disease
prevention measures, we can provide professional advice, but this question is beyond our expertise.
[TEXT_END]
[[Model Answer]]
[TEXT_START]
Regarding the question about genital herpes, based on the known information, I cannot provide a
relevant answer.
[TEXT_END]
[[Judgment Result]]
[SAME_ANSWER]
false
[SAME_ANSWER]
[REASON_START]
The model answer lacks the specific reasons and subsequent recommendations provided in the standard
answer.
[REASON_END]

Input/Output Example 4:
[[Standard Answer]]
[TEXT_START]
Based on the known information, the support status for the EasyMesh version of TL-XDR6010 is as
follows:
1. **TL-XDR6010 EasyMesh Version 1.0**: Supports EasyMesh 2.0 out of the box.
2. **TL-XDR6010 EasyMesh Version 2.0**: Requires upgrading to firmware version 1.0.9 Build 20240205
Rel.35518 or later to support EasyMesh 2.0.
[TEXT_END]
[[Model Answer]]
[TEXT_START]
Apologies, the smart assistant cannot answer this question yet.
The known information cannot address this question.
[TEXT_END]
[[Judgment Result]]
[SAME_ANSWER]
false
[SAME_ANSWER]
[REASON_START]
The model answer and standard answer are inconsistent in their statements.
[REASON_END]

Input/Output Example 5:
[[Standard Answer]]
[TEXT_START]
Thought: The user expressed an intent to query the parking location of a vehicle, which aligns with
the task information. However, the user did not provide any specific parameter information. According
to the task requirements, the first mandatory parameter, "license plate number," must be inquired
about first.

Final Answer: {"Parameters": {"License Plate Number":"", "Query Date":"", "Query Time Range":""},
"Reply": "May I ask for your license plate number? The license plate number is the official
registration number used to uniquely identify your vehicle, e.g., Yue B12345.", "Is Conversation
Over": "No"}
[TEXT_END]
[[Model Answer]]
[TEXT_START]
Thought: The user expressed an intent to query the parking location of a vehicle, which aligns with
the task information. However, the user did not provide any specific parameter information.
Therefore, the first mandatory parameter, "license plate number," must be inquired about first.
Final Answer: {"Parameters":{"License Plate Number":"","Query Date":"","Query Time
Range":""},"Reply":"May I ask for your license plate number? The license plate number is the official
registration number used to uniquely identify your vehicle, e.g., Yue B12345.","Is Conversation
Over":"No"}
[TEXT_END]
[[Judgment Result]]
[SAME_ANSWER]
true
[SAME_ANSWER]
[REASON_START]
The key points and Final Answer in the model answer are consistent with the standard answer, so the
answer consistency is true.
[REASON_END]

Now, given the standard answer and model answer below, please provide the judgment result for answer
consistency based on the examples and requirements above:
[[Standard Answer]]
[TEXT_START]
{raw_answer}
[TEXT_END]
[[Model Answer]]
[TEXT_START]
{model_answer}
[TEXT_END]

[[Judgment Result]]
```

