# OpenReview forum: "ASPD: Unlocking Adaptive Serial-Parallel Decoding by Exploring Intrinsic Parallelism in LLMs"
_ICLR.cc/2026/Conference — Submitted to ICLR 2026_

### Official Review · Reviewer_9wUP · 2025-10-14

**Soundness:** 3
**Presentation:** 2
**Contribution:** 3
**Rating:** 4
**Confidence:** 4

**Summary:**

The paper introduces ASPD, a method that uncovers and exploits intrinsic parallelism in AR LLM outputs by identifying semantically independent segments that can be decoded in parallel.
A dataset processing pipeline is also proposed.
Evaluation results show that ASPD can speed up inference with under a 1% drop in output quality across tasks.

**Strengths:**

1. The paper proposes a new direction for test time scaling, where the LLM can generate texts using its intrinsic ability.
2. The speedup comes with no additional compute cost.
3. The parallel data generation pipeline seems interesting.
4. It effectively overcomes the autoregressive bottleneck at the segment level.

**Weaknesses:**

1. The methodology for the dataset generation (Section 3.1) is not very clear to me. How do you rewrite the data into parallel and serial data? How is the verification done in detail? I feel Section 3.1 could be elaborated further with more clarity.
2. There are not much details on the inference engine, like how it handles batching and so on.
3. The evaluation does not compare with SOTA verifier-guided beam search methods.


Minor:
1. Figure 1 and 2 font sizes are too small to read.

**Questions:**

In addition to the weaknesses mentioned above, there are a few more questions:

1. What is the major difference between ASPD and Multiverse?
2. How does the inference engine maximize the inference efficiency?
3. What is the batch size used for inference in experiments?
4. What is the training cost?

---

> ### Author Response · Authors · 2025-11-20
> **Response to Reviewer 9wUP (1/4)**
>
> We sincerely thank the reviewers valuable and insightful comments. Our responses are as follows:
> ***
> **W1**:dataset generation detail
>
> **A1**:Thank you for your valuable feedback. We agree that Section 3.1 needs greater clarity, and we appreciate your suggestion to improve it.
> Due to space limitations, the detailed prompts for dataset generation were originally placed in Appendix 7, and implementation details were provided in the source code. As you rightly pointed out, this part is crucial. In the revised version, we have integrated these details into Section 3.1 for better readability and completeness. Please refer to our updated submission.
>
> 1. **Rewriting into parallel and serial data**:
> We use a parallel rewriting prompt to convert original answers into either parallel or serial formats. Given an original answer, we prompt an LLM to organize the potential parallel branches into labeled branches using special tags. Below is an example of the model’s output with parallel-structure markers:
>
> ```text
> To connect and use a Lenovo wireless mouse and keyboard, follow these steps:
> <branchgroup><branch title="Wireless Mouse">1. **Connect the Wireless Mouse**:
> - Ensure the mouse has sufficient battery power and turn it on.
> - Insert the receiver into the computer's USB port.
> - Press and hold the mouse scroll wheel while turning it to ON, hold for 3 seconds until the
> indicator light flashes, then bring it close to the receiver to complete pairing.</branch>
> <branch title="Wireless Keyboard">2. **Connect the Wireless Keyboard**:
> - Ensure the keyboard has sufficient battery power and turn it on.
> - Insert the receiver into the computer's USB port.
> - Simultaneously press the "F2," "F3," and "F4" keys on the keyboard, turn it to ON, release the
> keys, and press "3." The rapid flashing of the indicator light indicates successful pairing.</branch></branchgroup>
> By following these steps, you can successfully connect and use Lenovo's wireless mouse and keyboard.
> ```
>
> The `<branchgroup>` and `<branch>` tags explicitly mark parallel structures, enabling clear extraction and downstream processing. If the LLM’s output indicates that parallel rewriting is not possible, then that data item will remain in its original format (i.e., as serial data). It is worth noting that for each individual data item, we sample three LLM-rewritten responses to ensure diversity and improve the success rate of rewriting. Successfully rewritten samples are then passed to the next verification stage for further processing.
>
> 2. **Verification process**:
> For verification, we feed extracted branch content and context into a validation prompt, asking an LLM to judge whether branches are logically valid and independent. To enhance reliability and reduce model uncertainty, we apply majority voting across multiple judgments, taking the majority outcome as the final decision.
>
> We hope these clarifications address your concerns, and we thank you again for helping us improve the paper.

---

> ### Author Response · Authors · 2025-11-20
> **Response to Reviewer 9wUP (2/4)**
>
> **W2**: detail of inference engine  &  **Q2** How does the inference engine maximize the inference efficiency?
>
> **A**: Our inference engine optimizes efficiency by integrating our model architecture design and the parallelism-detection capability acquired post model training. It features a streamlined design, requiring only six additional special tokens  `<title>, </title>, <branch>, </branch>, <para>, </para>`for discriminative control in code. We elaborate on its workflow across three key dimensions:
>
>
> ### 1. Serial-to-Parallel Transition
> In serial decoding mode, when the model identifies upcoming parallelizable branches, it wraps each branch’s title in` <title></title> `to distinguish between branches. After all parallel titles are generated, the model outputs a <para> token—triggering the engine to switch to parallel decoding mode.
>
>
> ### 2. Parallel Decoding Mode
> Upon entering parallel mode:
> - The engine assigns position IDs (posIDs), attention masks, and parallel prefixes (`<branch>`$T_i$: , where $T_i$ denotes the branch’s title) to each parallel branch.
> - Attention masks ensure mutual invisibility between branches; standard causal masks are used within each branch.
> - Parallel branch posIDs are sequential relative to the preceding main branch, and posIDs across branches are synchronized at the same timestamp.
>
> When a single branch finishes decoding, the model generates `</branch>`; once all branches complete, the engine appends `</para>` to mark the end of parallel mode. Notably, the engine dynamically maintains branch position information—eliminating the need for padding tokens or extra KV-cache pools. Unlike PDOS [1] (which requires padding to align branch lengths), this avoids redundant tokens and ensures high token utilization.
>
>
> ### 3. Parallel-to-Serial Transition
> Upon returning to serial decoding, our continuous KV-cache (non-batch, non-multi-sequence) and the trained shared-posID mechanism for parallel branches allow direct reuse of parallel token KV-caches—no recomputation is needed for subsequent serial steps.
>
> In contrast:
> - SoT [2] re-prefills with context and points during the point expansion phase, slowing inference for long contexts;
> - APAR [3] discards parallel branch KV-caches, preventing serial decoding from leveraging branch information;
> - PDOS [1] reassigns posIDs to parallel tokens when merging, necessitating re-prefilling of their KV-caches.
>
>
>  References
>
> [1] Yijiong Yu. Accelerate parallelizable reasoning via parallel decoding within one sequence. arXiv preprint arXiv:2503.20533, 2025.
>
> [2] Ning, X., Lin, Z., Zhou, Z., Yang, H., and Wang, Y. Skeleton-of-thought: Large language models can do parallel decoding. arXiv preprint arXiv:2307.15337, 2023.
>
> [3] Liu, M., Zeng, A., Wang, B., Zhang, P., Tang, J., and Dong, Y. APAR: LLMs can do auto-parallel auto-regressive decoding. arXiv preprint arXiv:2401.06761, 2024.

---

> ### Author Response · Authors · 2025-11-20
> **Response to Reviewer 9wUP (3/4)**
>
> **W2**: how it handles batching & **Q3**:What is the batch size used for inference in experiments?
>
> **A**:Thank you for your question regarding the inference batch size in our experiments.
>
> In our experiments, we use a batch size of 1, which aligns with common practices in parallel decoding literature—consistent with works such as Speculative Decoding [1], PASTA[2], and Multiverse [3]. This choice serves two key purposes: first, it ensures better comparability of evaluation results across different parallel decoding methods; second, our approach is compatible with mainstream inference frameworks, enabling rapid adaptation.
>
> To further demonstrate scalability, we integrated our model and inference logic into SGLang [4] (a widely used mainstream inference framework) and conducted evaluations on a combined test set (merging MT-Bench, Vicuna-Bench, and RAG-Bench) under various batch sizes. The results are presented in the table below(conducted on 4 H20 GPUs):
>
> | Batch Size | Wall-Clock Time (s) | Speed-Up Ratio (vs. Batch Size 1) |
> |------------|---------------------|-----------------------------------|
> | 128        | 39.23               | 90.92%                            |
> | 64         | 45.39               | 89.49%                            |
> | 32         | 58.79               | 86.40%                            |
> | 16         | 87.95               | 79.65%                            |
> | 8          | 137.52              | 68.19%                            |
> | 4          | 193.39              | 55.26%                            |
> | 2          | 292.70              | 32.29%                            |
> | 1          | 432.36              | -                                 |
>
> As shown in the table, when our inference method is integrated into the SGLang framework, inference speed is further improved under multi-batch settings: the speed-up ratio (relative to batch size 1) increases significantly with larger batch sizes, confirming the strong scalability of our approach.
>
>
> ### References
> [1] Leviathan, Y., Kalman, M., and Matias, Y. Fast inference from transformers via speculative decoding. In *International Conference on Machine Learning*, 2023.
> [2] Tian Jin, Ellie Y Cheng, Zack Ankner, Nikunj Saunshi, Blake M Elias, Amir Yazdanbakhsh, Jonathan Ragan-Kelley, Suvinay Subramanian, and Michael Carbin. Learning to keep a promise: Scaling language model decoding parallelism with learned asynchronous decoding. arXiv preprint arXiv:2502.11517, 2025.
> [3] Xinyu Yang, Yuwei An, Hongyi Liu, Tianqi Chen, and Beidi Chen. Multiverse: Your language models secretly decide how to parallelize and merge generation. arXiv preprint arXiv:2506.09991, 2025.
> [4] Lianmin Zheng, Liangsheng Yin, Zhiqiang Xie, Chuyue Livia Sun, Jeff Huang, Cody Hao Yu, Shiyi Cao, Christos Kozyrakis, Ion Stoica, Joseph E Gonzalez, et al. Sglang: Efficient execution of structured language model programs. *Advances in Neural Information Processing Systems*, 37:62557–62583, 2024.

---

> ### Author Response · Authors · 2025-11-20
> **Response to Reviewer 9wUP (4/4)**
>
> **W3**:compare with SOTA verifier-guided beam search methods.
>
> **A3**: Thank you for your valuable suggestion regarding comparisons with SOTA verifier-guided beam search methods.
>
> Verifier-guided beam search falls under the perspective of test-time scaling, which focuses on parallelizing **multiple solutions to the same problem**—a direction aligned with works like Parallel-R1 [1] and ParaThinker [2]. In contrast, our ASPD adopts an acceleration-centric perspective: we achieve parallelism by **decomposing the internal steps of a single solution to the same problem**, with the core goal of boosting inference speed.
>
> Your suggestion provides an excellent insight: integrating ASPD with such test-time scaling-oriented parallel methods (e.g., verifier-guided beam search) could enable step-decomposition parallelization *within each complete solution path*. This hybrid approach is expected to yield a more optimal "high-quality + high-speed" parallel solution.
>
> References
>
> [1] Tong Zheng, Hongming Zhang, Wenhao Yu, Xiaoyang Wang, Xinyu Yang, Runpeng Dai, Rui Liu, Huiwen Bao, Chengsong Huang, Heng Huang, et al. Parallel-R1: Towards parallel thinking via reinforcement learning. arXiv preprint arXiv:2509.07980, 2025.
> [2] Hao Wen, Yifan Su, Feifei Zhang, Yunxin Liu, Yunhao Liu, Ya-Qin Zhang, and Yuanchun Li. ParaThinker: Native parallel thinking as a new paradigm to scale LLM test-time compute. arXiv preprint arXiv:2509.04475, 2025.
> ***
>
> Minor: Figure 1 and 2 font sizes are too small to read.
>
> **A**:Thank you for your valuable feedback. We appreciate your suggestion and have increased the font sizes in Figures 1 and 2 in the refined version to enhance readability.
> ***
>
> **Q1**: Difference between ASPD and Multiverse?
>
> **A1**:Thank you for your question regarding the core differences between ASPD and Multiverse.
>
> Multiverse[1] is an excellent concurrent work—its arXiv submission was about 2 months earlier than ours, while its formal acceptance came after our ICLR submission. Although ASPD and Multiverse share common ground in exploring transformer parallel decoding, three key differences distinguish our approaches:
>
> ### 1. Core Parallelism Source & Data Pipeline
> ASPD focuses on **mining and leveraging the intrinsic parallelism inherently existing in the outputs of vanilla autoregressive (AR) models**. Our data pipeline is non-intrusive: it does not alter the original response distribution of AR models, relying solely on semantic-level selection via LLMs rather than rule-based modifications.
> In contrast, Multiverse is primarily tailored to mathematical reasoning tasks . Its design and optimization are more task-specific, which limits its adaptability beyond math domains.
>
>
> ### 2. Task Generalization
> Benefiting from our focus on intrinsic parallelism (a universal trait across AR model responses), ASPD exhibits strong generalization across diverse tasks—including STEM, RolePlay, Reasoning, RAG, and Information Extraction—while maintaining speedup advantages.
> Multiverse, by contrast, concentrates exclusively on mathematical tasks, lacking such cross-domain adaptability.
>
> ### 3. Parallel Decoding Implementation
> ASPD realizes parallel decoding through a **one-sequence paradigm**: both serial and parallel decoding processes are completed within a single sequence. This design ensures continuous KV-cache storage and direct reuse of parallel branch KVs in subsequent serial steps, eliminating redundant computation. However, Multiverse employs radix attention for MapReduce to merge parallel branches into the main branch.
>
>  Reference
>
> [1] Xinyu Yang, Yuwei An, Hongyi Liu, Tianqi Chen, and Beidi Chen. Multiverse: Your language models secretly decide how to parallelize and merge generation. arXiv preprint arXiv:2506.09991, 2025b.
>
> We hope this clarification articulates the distinguishing features of our work.
> ***
>
> **Q3**: quantitative analysis of training overhead
>
> **A3**: Thank you very much for your valuable comment. ASPD modifies the attention mask and position settings during training, but this does not introduce additional training time overhead. Below, we provide a quantitative comparison of Sequential Fine-Tuned (Seq) and ASPD models under the same training configuration, software, and hardware environment(8 H20 GPUs). For more details on experimental parameters, please refer to Section 4.1 Implementation Details:
>
> | Model | Training Time      |
> |-------|-------------------|
> | V-Seq   | 6 hours 24 minutes |
> | V-ASPD  | 6 hours 17 minutes |
>
> As shown in the above table, the training time for ASPD is similar to that of Seq. This indicates that our ASPD framework does not introduce significant training overhead, thus enabling parallel decoding for faster inference without requiring extra training resources.

---

### Official Review · Reviewer_MYWg · 2025-10-30

**Soundness:** 2
**Presentation:** 2
**Contribution:** 3
**Rating:** 4
**Confidence:** 4

**Summary:**

The paper addresses the inference‑latency bottleneck in current LLMs due to strictly sequential autoregressive decoding. The authors observe that many generated responses contain intrinsic parallelism — segments that can be produced independently without breaking coherence. Motivated by this, they propose Adaptive Serial‑Parallel Decoding (ASPD), which addresses two core
challenges: automated construction of parallelizable data and an efficient parallel decoding mechanism. Experiments show that this method provides significant improvements in both effectiveness and efficiency compared to existing approaches.

**Strengths:**

Parallel decoding is a promising technique for inference acceleration. While previous works mainly focus on token-level parallel decoding (i.e., decode multiple tokens simultaneously), this paper leverages the intrinsic parallelism in LLMs. This is a good motivation.

Speed gains across diverse domains and models, with minimal trade‑off in output quality.

**Weaknesses:**

see questions

**Questions:**

The paper says: Tokens in the main branch maintain absolute positions in the flattened sequence, while parallel branches synchronize their position encodings at each timestamp. (line 269). Does this mean that tokens in parallel branches have two position ids: one for the main branch and the other for the parallel branches? If so, parallel tokens will recompute KVs when they are flattened and merged into the main branch, which introduces extra cost. If not, the position ids in the main branch are problematic.

What is the average and variance of parallel branch lengths? If branches have very different lengths, the decoding will be blocked by the longest branch.

---

> ### Author Response · Authors · 2025-11-20
> **Response to Reviewer MYWg (1/2)**
>
> We sincerely thank the reviewers valuable and insightful comments. Our responses are as follows:
> ***
>
> **Q1**: tokens in parallel branches have two position ids？ Does it need recomputing KVs when they are merged into the main branch？
>
> **A1**:Thank you for your thoughtful question regarding the position IDs of tokens and the handling of KV-cache in our framework.
>
> We would like to clarify that this is not the case—tokens (whether in the main branch or parallel branches) only have one set of position IDs. A key advantage of ASPD is that when parallel branches are merged into the main branch, we do not need to recompute the KV-cache of the already decoded parallel tokens.
>
> To illustrate, suppose the preceding main branch tokens have position IDs 1, 2, and 3. If the model identifies three parallel branches (A, B, C) with lengths 3, 2, and 3 respectively, their position encodings are assigned as follows: A: 4, 5, 6; B: 4, 5; C: 4, 5, 6. After parallel decoding completes, no new position IDs are reassigned to these tokens, allowing their KV-cache to be directly reused—this design is a core feature validated through our model training.
>
> In contrast, existing methods suffer from the limitations you noted:
> PDOS [1] uses two sets of position IDs for parallel tokens (one during parallel decoding and another when merged into serial decoding), thus requiring KV-cache recomputation for merged tokens;
> PASTA [2] necessitates pre-predicting the length of parallel branches;
> APAR [3] discards the KV-cache of parallel branches, which restricts the quality of subsequent serial decoding;
> SoT [4] performs re-prefilling with context and points during the point expansion phase—this prefilling slows down inference when the context is long.
>
> References
>
> [1] Yijiong Yu. Accelerate parallelizable reasoning via parallel decoding within one sequence. arXiv preprint arXiv:2503.20533, 2025.
>
> [2] Tian Jin, Ellie Y Cheng, Zack Ankner, Nikunj Saunshi, Blake M Elias, Amir Yazdanbakhsh, Jonathan Ragan-Kelley, Suvinay Subramanian, and Michael Carbin. Learning to keep a promise: Scaling language model decoding parallelism with learned asynchronous decoding. arXiv preprint arXiv:2502.11517, 2025.
>
> [3] Liu, M., Zeng, A., Wang, B., Zhang, P., Tang, J., and Dong, Y. APAR: LLMs can do auto-parallel auto-regressive decoding. arXiv preprint arXiv:2401.06761, 2024.
>
> [4] Ning, X., Lin, Z., Zhou, Z., Yang, H., and Wang, Y. Skeleton-of-thought: Large language models can do parallel decoding. arXiv preprint arXiv:2307.15337, 2023.

---

> > ### Comment · Reviewer_MYWg · 2025-11-26
> >
> > Thanks for your reply. But my concern still exists.
> >
> > In your case, three parallel branches (A, B, C) have lengths 3, 2, and 3. The Pids are: A: 4, 5, 6; B: 4, 5; C: 4, 5, 6. After parallel decoding completes, if reusing KVs, the position ids is like "1,2,3,4,5,6,4,5,4,5,6,7,8,...". This looks weird as the same position id is assigned to multiple tokens.

---

> > > ### Author Response · Authors · 2025-11-26
> > > **Response to Reviewer MYWg**
> > >
> > > Dear Reviewer MYWg, Thank you for your timely and valuable feedback.
> > >
> > > From the perspective of the merged main branch:
> > > ```
> > >               A4, A5, A6,
> > >
> > > M1, M2, M3,   B4, B5,       M7,M8,...
> > >
> > >               C4, C5, C6,
> > > ```
> > > Position ids of different parallel branches are identical at the same time step, which would prevent the model from understanding the sequence **if it were not properly trained. However, we maintain strict consistency between training and inference, enabling the model to adapt to this positional ids scheme**(as in our previous reply: this design is a core feature validated through our model training).
> > >
> > > This is precisely why our method does not require re-computing the kvcache for parallel tokens. Our concurrent work, Multiverse[1], also adopts the same positional ids paradigm.
> > > In the prior work, PASTA[2], they predict positional ids for parallel branches. Due to prediction inaccuracies, they also encounter cases where different tokens share identical positional ids.
> > >
> > > Thank you again for your response. I hope my explanations have addressed your concerns.
> > >
> > > References
> > >
> > > [1] Xinyu Yang, Yuwei An, Hongyi Liu, Tianqi Chen, and Beidi Chen. Multiverse: Your language models secretly decide how to parallelize and merge generation. arXiv preprint arXiv:2506.09991, 2025b.
> > >
> > > [2] Tian Jin, Ellie Y Cheng, Zack Ankner, Nikunj Saunshi, Blake M Elias, Amir Yazdanbakhsh, Jonathan Ragan-Kelley, Suvinay Subramanian, and Michael Carbin. Learning to keep a promise: Scaling language model decoding parallelism with learned asynchronous decoding. arXiv preprint arXiv:2502.11517, 2025.

---

> ### Author Response · Authors · 2025-11-20
> **Response to Reviewer MYWg (2/2)**
>
> **Q2.1**: What is the average and variance of parallel branch lengths?
>
> **A2.1**: We have calculated the average branch lengths and their standard deviations for both our training and test sets. For different parallel branches within the same response, due to the correlation in overarching themes and structures, the differences in their lengths are not particularly significant.
> The results are summarized in the following table:
>
> | Type | Dataset  | Avg Branch Length (Mean ± Std) |
> |-------------|---------------|--------------------------------|
> | Train Set    | ShareGPT Vicuna  | 100.51 ± 28.69                 |
> | Test Set     | MT Bench       | 114.78 ± 32.32                 |
> | Test Set     | Vicuna Bench   | 104.82 ± 27.54                 |
> | Test Set     | RAG Bench      | 86.63 ± 29.59                  |
>
> As shown the table, both training and test sets exhibit average parallel branch lengths close to 100, with a standard deviation around 30. This indicates that our model learns the intrinsic parallelism within the training data, and the branch length distributions remain consistent during inference.
> ***
>
> **Q2.2**: If branches have very different lengths, the decoding will be blocked by the longest branch.
>
> **A2.2**: This issue does exist in our parallel framework, and it is also a common challenge across many parallel decoding works—such as Multiverse [1], PASTA [2], APR [3], and PDOS [4]. Although the decoding of subsequent sequential tokens must wait for the longest parallel branch to complete, if critical information resides in these parallel branches and the subsequent sequential decoding proceeds without waiting for this information, the generation quality of the sequential sequence would often be compromised.
> This scenario represents an inherent trade-off in parallel decoding frameworks, and we therefore prioritize ensuring the quality of generated responses over pursuing extreme decoding speed. It is precisely this choice that allows us to achieve nearly lossless performance across multiple benchmarks while maintaining acceleration gains.
>
> Your suggestion is highly valuable. If a semantic-aware module could be integrated here—enabling the main branch to adaptively decide whether to wait for a specific parallel branch—it would optimize decoding speed to a certain extent. We plan to investigate this direction in our future work.
> ***
>
> References:
>
> [1] Xinyu Yang, Yuwei An, Hongyi Liu, Tianqi Chen, and Beidi Chen. Multiverse: Your language models secretly decide how to parallelize and merge generation. arXiv preprint arXiv:2506.09991, 2025b.
>
> [2] Tian Jin, Ellie Y Cheng, Zack Ankner, Nikunj Saunshi, Blake M Elias, Amir Yazdanbakhsh, Jonathan Ragan-Kelley, Suvinay Subramanian, and Michael Carbin. Learning to keep a promise: Scaling language model decoding parallelism with learned asynchronous decoding. arXiv preprint arXiv:2502.11517, 2025.
>
> [3] Jiayi Pan, Xiuyu Li, Long Lian, Charlie Snell, Yifei Zhou, Adam Yala, Trevor Darrell, Kurt Keutzer, and Alane Suhr. Learning adaptive parallel reasoning with language models. arXiv preprint arXiv: 2504.15466, 2025.
>
> [4] Yijiong Yu. Accelerate parallelizable reasoning via parallel decoding within one sequence. arXiv
> preprint arXiv:2503.20533, 2025.

---

### Official Review · Reviewer_aASG · 2025-10-30

**Soundness:** 2
**Presentation:** 4
**Contribution:** 3
**Rating:** 4
**Confidence:** 4

**Summary:**

The paper presents ASPD, a methodology for enabling parallel decoding. The method enables reusable KV cache and maintains ground truth position IDs during parallel decoding, with the capability to resume sequential generation mode after parallel generation mode. The work evaluates ASPD, showing it achieves increase in tokens/sec while maintaining the quality of sequential generation.

**Strengths:**

- ASPD enables parallel decoding while addressing the weaknesses of previous work (no sequential decoding after parallelizing in APAR; approximated position IDs disrupting position continuity in Pasta)
- The paper is generally well written and easy to understand, which the figures giving a very clear overview of the methodology and of differences with previous works.
- The experiments show that ASPD achieves the greatest tokens/sec and highest quality compared to APAR, SOT, and sequential across three benchmarks, demonstrating that ASPD does enable more tokens generated at time.

**Weaknesses:**

- The paper does not present the wall clock latency speedup of the different methods, but only tokens/sec and other efficiency metrics which do not account for actual system overheads to the methodology. As a speed-oriented parallelization method, wall clock speedup is an important evaluation metric.
- It seems that the main difference between ASPD and Pasta is that in ASPD the position ID is maintained as if the tokens generated in parallel were actually sequential (i.e. ground truth position IDs) while Pasta uses model predictions to compute the position IDs, which makes Pasta an important baseline. However, the evaluation doesn't compare against PASTA as a parallel decoding baseline in Figure 4, but only ablate the data pipeline methodology used in Pasta.

Minor comment:
- The colored grid lines on Figure 4 makes it difficult to read.

**Questions:**

Please address the above concerns.

---

> ### Author Response · Authors · 2025-11-20
> **Response to Reviewer aASG**
>
> We sincerely thank the reviewers valuable and insightful comments. Our responses are as follows:
> ***
>
> **Q1**: wall clock latency speedup of the different methonds.
>
> **A1**:Thank you for your valuable comment. A key advantage of our parallel method is that **no recomputation of already decoded tokens occurs when merging parallel branches into the main branch**, thus avoiding additional system overhead.
> Following your suggestion, we have supplemented wall clock latency (WCL) speedup evaluations across benchmarks, and the results are consistent with the tokens/sec metrics we previously reported. The detailed data (including both WCL speedup and generation quality scores) is presented in the table below:
>
> | Model    | MT Bench - WCL (Speedup) | MT Bench - Score | Vicuna Bench - WCL (Speedup) | Vicuna Bench - Score | RAG Bench - WCL (Speedup) | RAG Bench - Score |
> |----------|---------------------------|------------------|-------------------------------|-----------------------|----------------------------|-------------------|
> | V-Seq    | 15.5                      | 5.59             | 17.64                         | 7.70                  | 9.67                       | 8.29              |
> | V-APAR*  | 10.37 (33.10%)            | 5.38             | 12.42 (29.59%)                | 7.62                  | 7.41 (23.37%)              | 8.04              |
> | V-ASPD   | 8.54 (44.90%)             | 5.59             | 11.66 (33.90%)                | 7.74                  | 6.03 (37.64%)              | 8.21              |
>
> As shown, our ASPD achieves the highest WCL speedup (44.90% on MT Bench, 33.90% on Vicuna Bench, and 37.64% on RAG Bench) compared to the sequential baseline (V-Seq) and the parallel baseline (V-APAR*). Notably, ASPD maintains generation quality comparable to V-Seq (e.g., 5.59 vs. 5.59 on MT Bench) while delivering superior speedup—further confirming the effectiveness of our method in balancing latency reduction and output quality.
> ***
>
> **Q2.1**:Difference between ASPD and PASTA.
>
> **A2.1**:Thank you for your insightful comment, which helps us further clarify the distinctions between our work and PASTA.
> As you have noted, ASPD and PASTA differ in their handling of positional encoding for parallel-generated tokens. PASTA first predicts the length of each parallel branch and pre-allocates fixed positional encoding intervals for individual branches. When the actual decoding length is inconsistent with the prediction, it introduces comprehension conflicts for the model, as illustrated in Figure 2 (b) of our paper. In contrast, ASPD enables different parallel branches to share the same positional encoding at the same time. By training the model to adapt to this pattern, the kv-cache of parallel branches during inference can be directly reused in subsequent serial branches while preserving the model’s performance.
>
> Beyond positional encoding, ASPD also differs distinctly from PASTA in the following aspects:
>
> 1. Data Processing Pipeline: ASPD aims to exploit the inherent parallelism in autoregressive (AR) model outputs and adopts a non-intrusive design that makes no modifications to the AR model’s responses. PASTA, however, employs an intrusive data transformation pipeline that may alter the original data.
>
> 2. Decoding Mode: Benefiting from our model architecture design, the kv-cache of ASPD’s serial-parallel branches remains contiguous during decoding. Parallel decoding is performed simultaneously based on the actual number of branches, with the entire process confined within a single sequence—thus avoiding additional overhead when switching between serial and parallel decoding. PASTA utilizes a fixed-size kv-cache pool and achieves parallelism via batch decoding, which supports up to 4 parallel branches by default. This introduces extra memory copying and computation when switching between the main branch and parallel branches. Additionally, the fixed size of the kv-cache pool leads to low GPU memory utilization.
> ***
>
> **Q2.2**:the evaluation doesn't compare against PASTA as a baseline
>
> **A2.2**:Notably, PASTA does not provide open-source code, pre-trained models, or an inference engine. However, its paper publicly releases the prompt for data transformation and the implementation details of positional encoding. Given this constraint, we conducted ablation studies to compare our approach with PASTA across two key dimensions: the data pipeline (presented in Table 6) and positional encoding (posid, shown in Table 8).
> We hope this clarifies our experimental design regarding PASTA.
> ***
>
> **Q3**: The colored grid lines on Figure 4 makes it difficult to read.
>
> **A3**:Thank you for your valuable feedback. We have revised Figure 4 in the revised version of the paper to enhance its readability by removing the colored grid lines.

---

> > ### Comment · Reviewer_aASG · 2025-11-25
> >
> > Thank you for the responses.
> >
> > Q1: My concern is addressed, thank you.
> >
> > Q2.1: Thank you for the clarification. I would like to point out that PASTA does not utilize a fixed size kv-cache pool. It uses a contiguous interleaving kv-cache layout that supports any number of parallel branches and does not introduce any extra memory copying and computation when switching between the main branch and parallel branches. As such, maintaining a contiguous kv-cache is not a novel contribution. I acknowledge that maintaining position IDs is novel and so is introducing the independence verification in the data processing pipeline, but also want to correct the mischaracterization of PASTA and the over-claim of ASPD's contributions in your response.
> >
> > Q2.2: Thank you for the clarification. However, as noted, PASTA predicts the length of the branch and pre-allocates positional encoding based on that length. This is different from the ablation performed in this work. According to the paper draft, the `Fixed` ablation pre-allocates fixed-length intervals, not predicted length intervals. An important baseline/ablation that is missing is to use predicted length-intervals to allocate the position encoding, since that is the only difference between PASTA and ASPD besides the data pipeline (which I acknowledge the data pipeline ablation is adequate). As is, the evaluations/ablations do not answer whether it is important to maintain the positional encodings or if LLM can accurately predict length interval so the predicted position encodings perform well already.
> >
> > Q3: Thank you.

---

> ### Author Response · Authors · 2025-11-26
> **Response to Reviewer aASG**
>
> **A2.1**:
>
> Dear Reviewer aASG, We sincerely appreciate your correction. Due to the unavailability of open-source code for PASTA's[1] method, there was a deviation in our previous understanding. We acknowledge that PASTA utilizes a continuous kv-cache to maintain efficient parallel decoding.
>
> **We also thank you for recognizing the novelty of ASPD in exploring the inherent parallelism of data and its parallel position encoding**. As you correctly pointed out, the position encoding of parallel branches is a crucial aspect that distinguishes ASPD from PASTA. We will conduct more detailed ablation experiments on the Predict-based position encoding paradigm in PASTA in our response to A2.2.
> ***
>
> **A2.2**:
>
> Thank you very much for your valuable comments. More specifically,**PASTA mentions in the original paper**: "If the predicted number of tokens for the ```<async>``` block does not match the true number of tokens generated, this can lead to errors as the position IDs after synchronization will either not increase monotonically (predicted too few) or contain a gap (predicted too many)."
>
> To address this issue, PASTA explored four approaches in their work:
> 1. Oracle: Use the ground truth length of each ```<async>``` block
> 2. Fixed-length: Assume each ```<async>``` block has a fixed length
> 3. Predict-1x: Precisely predict the ```<async>``` token length
> 4. Predict-10x: Predict the token length as a multiple of ten for coarser granularity (achieving the best performance)
>
> Among these, Oracle serves as an ideal scenario where the predicted length perfectly matches the actual decoding length (which is impractical in real-world applications, as noted by the authors that this is merely a reference experiment). However, its actual performance is inferior to that of Predict-10x. Additionally, the "Predict" methods introduce a hyperparameter for the scaling factor, and the choice of '10' in the paper lacks specific justification.
>
> Based on these two considerations, we did not adopt the Predict methods in our previous work and instead used Fixed-length.
>
> As you correctly pointed out, further ablation studies on the Predict-10x setting are necessary. We conducted ablation experiments on PASTA's Predict-10x (with experimental settings fully consistent with Section 4.4), and the results are presented below:
>
> | Pos-Method   | TPS    | Score  |
> |--------------|--------|--------|
> | Fixed        | 59.11  | 6.09   |
> | Pred-10x     | 72.15  | 6.75   |
> | ASPD (ours)  | 104.21 | 7.64   |
>
> As shown in the results, the approach of predicting the length of parallel branches to assign position IDs outperforms the Fixed-length method in both speed and quality, but is still inferior to the Same scheme of ASPD. **Specifically, compared to Pred-10x, ASPD achieves a speed improvement of 44.43% and a quality improvement of 13.19%.**
>
> Furthermore, to evaluate the model's ability to predict the length of parallel branches, we statistically analyzed the deviation between the predicted length and the actual decoding length of parallel branches during inference. The calculation method is: [(Actual Length - Predicted Length) / Actual Length] × 100%.
>
> **The average prediction error rate is 24.29%, indicating that the model still faces considerable challenges in predicting its own generation length.**
>
> Based on the above ablation studies, we conclude that ASPD's position encoding scheme outperforms the Predict methods in both speed and quality. Thank you again for your insightful feedback. We will add the Predict position encoding method to Table 4: PosId in the revised version. We hope these experiments address your concerns. Thank you.
> ***
>
> References
>
> [1] Tian Jin, Ellie Y Cheng, Zack Ankner, Nikunj Saunshi, Blake M Elias, Amir Yazdanbakhsh, Jonathan Ragan-Kelley, Suvinay Subramanian, and Michael Carbin. Learning to keep a promise: Scaling language model decoding parallelism with learned asynchronous decoding. arXiv preprint arXiv:2502.11517, 2025.

---

### Official Review · Reviewer_Utbc · 2025-11-01

**Soundness:** 3
**Presentation:** 3
**Contribution:** 3
**Rating:** 6
**Confidence:** 2

**Summary:**

This paper introduces ASPD (Adaptive Serial-Parallel Decoding) to accelerate LLM inference by exploiting "intrinsic parallelism" in responses. Instead of pure autoregressive decoding, it identifies parallelizable structures via an automated, non-invasive data pipeline. A hybrid decoding engine then adaptively switches between serial and parallel generation, crucially maintaining and reusing the KV cache across modes. This approach achieved significant speedup up to 3.10x (1.82x avg) on Vicuna Bench while preserving generation quality with less than 1% degradation.

**Strengths:**

- The paper tackles an interesting aspect of the LLM parallelism. And the found intrinsic parallelism such as lists are interesting.

- The experiments are comprehensive and thorough, covering different reasoning tasks such as STEM, roleplay, reasoning, and extraction tasks.

**Weaknesses:**

- Speedups for certain tasks such as mathematics reasoning are limited. For example, the speedup on MATH500 is 1.17x, much lower than the 1.82x achieved on Vicuna Bench.

- The method is dependent on task structure. Mathematical reasoning, for instance, involves "strong inter-step dependencies" and "step-by-step deductions," which naturally reduces the opportunities for parallelization.

- The training overhead seems to be missing. What are the training overhead and how long does it take? Consider a quantitative analysis.

Miscellaneous
- Line 277 end: should be ``<branch>T_i:"

**Questions:**

See weakness.

---

> ### Author Response · Authors · 2025-11-20
> **Response to Reviewer Utbc**
>
> We sincerely thank the reviewers valuable and insightful comments. Our responses are as follows:
> ***
>
> **Q1**:Speedups for certain tasks such as mathematics reasoning are limited. & **Q2**: inter-step dependencies tasks reduces the opportunities for parallelization.
>
> **A1&A2**:Thank you for your comment on the limited speedups in tasks has strong inter-step dependencies, like mathematical reasoning.
> Our approach leverages the intrinsic parallelism of autoregressive (AR) model outputs for parallel decoding. As shown in Figure 1, mathematical tasks have the lowest Degree of Parallelism (DP, 30.6%), which directly leads to the modest 1.17× speedup on MATH500 (vs. 1.82× on Vicuna Bench). This confirms our method does not alter the model’s original response distribution—instead, it merely harnesses task-specific intrinsic parallelism, which varies by domain. For tasks like AI-Search, RAG, and general QA, ASPD still identifies abundant parallel scenarios to deliver significant speedups.
> As noted in our Future Work, we will use pre-defined parallelism metrics (e.g., PPD, DP, ABN) with reinforcement learning (RL) to adjust the model’s response distribution, further enhancing parallelism across all tasks.
> We appreciate your valuable feedback.
> ***
>
> **Q3**: quantitative analysis of training overhead
>
> **A3**: Thank you very much for your valuable comment. ASPD modifies the attention mask and position settings during training, but this does not introduce additional training time overhead. Below, we provide a quantitative comparison of Sequential Fine-Tuned (Seq) and ASPD models under the same training configuration, software, and hardware environment(8 H20 GPUs). For more details on experimental parameters, please refer to Section 4.1 Implementation Details:
>
> | Model | Training Time      |
> |-------|-------------------|
> | V-Seq   | 6 hours 24 minutes |
> | V-ASPD  | 6 hours 17 minutes |
>
> As shown in the above table, the training time for ASPD is similar to that of Seq. This indicates that our ASPD framework does not introduce significant training overhead, thus enabling parallel decoding for faster inference without requiring extra training resources.
> ***
>
> **Q4**: Line277 typo
>
> **A4**: Thank you for pointing out this typo. We have corrected it to $T_i$ in the revised version.

---

### Author Response · Authors · 2025-11-20
**Response to All Reviewers**

We sincerely thank the reviewers for their careful review and valuable comments. Since some questions relate to our motivation, we first summarize it briefly and then provide targeted responses.

Our core motivation stems from the observation in Figure 1 that **autoregressive (AR) models exhibit inherent parallelism in their outputs across various tasks**. We aim to exploit this intrinsic parallelism, achieved through:
- **Data Pipeline**: A non-intrusive data curation process that preserves the original outputs of AR models. Instead of rule-based filtering, we rely on LLM-enabled semantic selection with multi-layer validation to ensure semantic independence among parallel branches.
- **Model Architecture**: We apply masking between parallel branches, with their position encodings being continuous relative to the preceding main branch. This enables parallel decoding within a *single sequence*, unlike batch-based (SoT [1]) or multi-sequence (PASTA [2]) approaches. From a parallel branch’s perspective, decoding follows the same mechanism as vanilla AR models, thus preserving their original capabilities.
- **Inference Efficiency**: Thanks to the above design and training, both serial and parallel decoding occur within a single sequence with continuous memory. KV-caches of parallel branches are directly reusable by subsequent main branches without recomputation.

Finally, our focus on intrinsic parallelism grants strong generalization: we are the first parallel decoding method to explore and achieve a speed-quality balance across multiple domains(STEM, RolePlay, Reasoning, RAG, Extraction Task), and we appreciate that all reviewers have recognized this strength.


 **To Reviewer Utbc**
(Q1: ASPD has low speedup on math; Q2: Low speedup on dependent tasks)
Since we do not alter the output distribution of the original AR model, our parallel model behaves identically to the original AR model in step-by-step scenarios. As shown in Figure 1, math tasks have the lowest parallelism, which explains the observed speedup patterns.


**To Reviewer MYWg**
(Q1: Whether tokens have two sets of position encodings requiring recomputation)
Tokens (whether in the main branch or parallel branches) only have one set of position IDs.Owing to our architecture, transitions between serial and parallel branches incur no performance or quality penalty. Thus, there is no recomputation of parallel tokens, maximizing inference efficiency.


**References**
[1] Ning, X., Lin, Z., Zhou, Z., Yang, H., and Wang, Y. Skeleton-of-thought: Large language models can do parallel decoding. arXiv preprint arXiv:2307.15337, 2023.
[2] Tian Jin, Ellie Y Cheng, Zack Ankner, Nikunj Saunshi, Blake M Elias, Amir Yazdanbakhsh, Jonathan Ragan-Kelley, Suvinay Subramanian, and Michael Carbin. Learning to keep a promise: Scaling language model decoding parallelism with learned asynchronous decoding. arXiv preprint arXiv:2502.11517, 2025.

---

### Author Response · Authors · 2025-12-01
**Summary To AC - Difference between Other Parallel Decoding Works and Our ASPD**

Dear Area chairs,

Thank you for your efforts during the paper review process. We have conducted a comprehensive analysis and summary of our paper and the reviewers' feedback to help you quickly understand the reviewers' concerns and the work we have done in our paper:

## Difference between Other Parallel Decoding Works and Our ASPD
***

### PASTA v.s. Our ASPD @ `Reviewer aASG W2`
Thanks to Reviewer aASG for acknowledging the novelty of ASPD that `maintaining position IDs is novel and so is introducing the independence verification in the data processing pipeline`.As Reviewer aASG pointed out that `the position encoding of parallel branches is a crucial aspect that distinguishes ASPD from PASTA`, we conduct more detailed ablation experiments on the Predict-based position encoding paradigm in PASTA below:

**PASTA mentions in the original paper**: "If the predicted number of tokens for the ```<async>``` block does not match the true number of tokens generated, this can lead to errors as the position IDs after synchronization will either not increase monotonically (predicted too few) or contain a gap (predicted too many)". To address this issue, PASTA use Predict-10x (Predict the token length as a multiple of ten for coarser granularity (achieving the best performance)) as the final choice for predicting position encoding paradigm.

We conducted ablation experiments on PASTA's Predict-10x (with experimental settings fully consistent with Section 4.4), and the results are presented below:

| Pos-Method   | TPS    | Score  |
|--------------|--------|--------|
| Fixed        | 59.11  | 6.09   |
| Pred-10x     | 72.15  | 6.75   |
| ASPD (ours)  | 104.21 | 7.64   |

As shown in the results, the approach of predicting the length of parallel branches to assign position IDs outperforms the Fixed-length method in both speed and quality, but is still inferior to the Same scheme of ASPD. **Specifically, compared to Pred-10x, ASPD achieves a speed improvement of 44.43% and a quality improvement of 13.19%.**

Furthermore, to evaluate the model's ability to predict the length of parallel branches, we statistically analyzed the deviation between the predicted length and the actual decoding length of parallel branches during inference. The calculation method is: [(Actual Length - Predicted Length) / Actual Length] × 100%.

**The average prediction error rate is 24.29%, indicating that the model still faces considerable challenges in predicting its own generation length.**

Based on the above ablation studies, we conclude that ASPD's position encoding scheme outperforms the Predict methods in both speed and quality. Thank Reviewer aASG again for your insightful feedback. We will add the Predict position encoding method to Table 4: PosId in the revised version. We hope these experiments address Reviewer aASG's concerns.

***

### Multiverse v.s. Our ASPD @ `Reviewer 9wUP Q1`

**Multiverse[3] is an excellent concurrent work—its arXiv submission was about 2 months earlier than ours, while its formal acceptance came after our ICLR submission.**  Although ASPD and Multiverse share common ground in exploring transformer parallel decoding, three key differences distinguish our approaches:

 1. Core Parallelism Source & Data Pipeline
ASPD focuses on **mining and leveraging the intrinsic parallelism inherently existing in the outputs of vanilla autoregressive (AR) models**. Our data pipeline is non-intrusive: it does not alter the original response distribution of AR models, relying solely on semantic-level selection via LLMs rather than rule-based modifications.
In contrast, Multiverse is primarily tailored to mathematical reasoning tasks . Its design and optimization are more task-specific, which limits its adaptability beyond math domains.


 2. Task Generalization
Benefiting from our focus on intrinsic parallelism (a universal trait across AR model responses), ASPD exhibits strong generalization across diverse tasks—including STEM, RolePlay, Reasoning, RAG, and Information Extraction—while maintaining speedup advantages.
Multiverse, by contrast, concentrates exclusively on mathematical tasks, lacking such cross-domain adaptability.

 3. Parallel Decoding Implementation
ASPD realizes parallel decoding through a one-sequence paradigm: both serial and parallel decoding processes are completed within a single sequence. This design ensures continuous KV-cache storage and direct reuse of parallel branch KVs in subsequent serial steps, eliminating redundant computation. However, Multiverse employs radix attention for MapReduce to merge parallel branches into the main branch.

---

### Author Response · Authors · 2025-12-01
**Summary To AC - Reviewers' Most Concerns**

## Reviewers' Most Concerns
***

### Inference Efficiency / Latency Issues

#### Questions
 - Reviewer aASG W1: `The paper does not present the wall clock latency speedup of the different methods.`
 - Reviewer 9wUP W2: `how it handles batching?`
 - Reviewer 9wUP Q3: `What is the batch size used for inference in experiments?`

#### Responses
##### To `Reviewer aASG W1:`
A key advantage of our parallel method is that **no recomputation of already decoded tokens occurs when merging parallel branches into the main branch**, thus avoiding additional system overhead.
Following your suggestion, we have supplemented wall clock latency (WCL) speedup evaluations across benchmarks, and the results are consistent with the tokens/sec metrics we previously reported. The detailed data (including both WCL speedup and generation quality scores) is presented in our `Response to Reviewer aASG A1`.

Our ASPD achieves the highest WCL speedup (44.90% on MT Bench, 33.90% on Vicuna Bench, and 37.64% on RAG Bench) compared to the sequential baseline (V-Seq) and the parallel baseline (V-APAR*). Notably, ASPD maintains generation quality comparable to V-Seq (e.g., 5.59 vs. 5.59 on MT Bench) while delivering superior speedup—further confirming the effectiveness of our method in balancing latency reduction and output quality.

##### To `Reviewer 9wUP W2` and `Reviewer 9wUP Q3`:
In our experiments, we use a batch size of 1, which aligns with common practices in parallel decoding literature—consistent with works such as Speculative Decoding [1], PASTA[2], and Multiverse [3]. This choice serves two key purposes: first, it ensures better comparability of evaluation results across different parallel decoding methods; second, our approach is compatible with mainstream inference frameworks, enabling rapid adaptation.

To further demonstrate scalability, we integrated our model and inference logic into SGLang [4]  and conducted evaluations on a combined test set (merging MT-Bench, Vicuna-Bench, and RAG-Bench) under various batch sizes. The results are presented in our `Response to Reviewer 9wUP (3/4)`. When our inference method is integrated into the SGLang framework, inference speed is further improved under multi-batch settings: the speed-up ratio (relative to batch size 1) increases significantly with larger batch sizes, confirming the strong scalability of our approach.

***

### Parallel Acceleration across Different Domain Tasks
#### Questions
 - Reviewer Utbc W1: `Speedups for certain tasks such as mathematics reasoning are limited. `
 - Reviewer Utbc W2: `The method is dependent on task structure. `

#### Responses
##### To `Reviewer Utbc W1` and `Reviewer Utbc W2`:

Our motivation is to leverage the intrinsic parallelism of autoregressive (AR) model outputs for parallel decoding. As shown in Figure 1, mathematical tasks have the lowest Degree of Parallelism (DP, 30.6%) in AR model's response, which directly leads to the modest 1.17× speedup on MATH500 (vs. 1.82× on Vicuna Bench). This confirms our method does not alter the model’s original response distribution—instead, it merely harnesses task-specific intrinsic parallelism, which varies by domain. This further accounts for why our method, differentiated from alternative approaches, exhibits robust generalization capabilities across a variety of tasks. Furthermore,  for tasks like AI-Search, RAG, and general QA, ASPD still identifies abundant parallel scenarios to deliver significant speedups.
***

### Training Cost
#### Questions
 - Reviewer Utbc W3: `The training overhead seems to be missing. `
 - Reviewer 9wUP Q4: `What is the training cost?`

#### Responses
##### To `Reviewer Utbc W3` and `Reviewer 9wUP Q4`:
| Model | Training Time      |
|-------|-------------------|
| V-Seq   | 6 hours 24 minutes |
| V-ASPD  | 6 hours 17 minutes |

As shown in the table, the training time for ASPD is close to that of Seq.
Detailed in our `Response to Reviewer Utbc A3`.

***

### References
[1] Leviathan, Y., Kalman, M., and Matias, Y. Fast inference from transformers via speculative decoding. In *International Conference on Machine Learning*, 2023.
[2] Tian Jin, Ellie Y Cheng, Zack Ankner, Nikunj Saunshi, Blake M Elias, Amir Yazdanbakhsh, Jonathan Ragan-Kelley, Suvinay Subramanian, and Michael Carbin. Learning to keep a promise: Scaling language model decoding parallelism with learned asynchronous decoding. arXiv preprint arXiv:2502.11517, 2025.
[3] Xinyu Yang, Yuwei An, Hongyi Liu, Tianqi Chen, and Beidi Chen. Multiverse: Your language models secretly decide how to parallelize and merge generation. arXiv preprint arXiv:2506.09991, 2025.
[4] Lianmin Zheng, Liangsheng Yin, Zhiqiang Xie, Chuyue Livia Sun, Jeff Huang, Cody Hao Yu, Shiyi Cao, Christos Kozyrakis, Ion Stoica, Joseph E Gonzalez, et al. Sglang: Efficient execution of structured language model programs. *Advances in Neural Information Processing Systems*, 37:62557–62583, 2024.

---

### Meta-Review · Area_Chair_urep · 2026-01-13

**Summary:**

This paper proposes ASPD, a method to accelerate LLM inference by exploiting intrinsic parallelism in autoregressive outputs. Strengths include an interesting observation about parallelizable structures in LLM outputs and a data pipeline for extracting such patterns. However, there are notable weaknesses: the speedups are highly task-dependent (only 1.17x on math vs 1.82x on Vicuna Bench), limiting practical applicability. The position ID scheme where parallel branches share the same IDs is unconventional and raises questions about model behavior (while it works through training, the theoretical justification is weak). The method overlaps significantly with prior work (PASTA), with the main novelty being the shared position ID scheme whose generality is unclear. The experimental comparisons initially lacked key baselines (PASTA's Pred-10x).

**Reviewer Concerns:**

Addressed: Wall-clock latency speedup was provided. PASTA Pred-10x comparison was added post-rebuttal. Batching scalability and training overhead were clarified. Outstanding: (1) The position ID scheme where multiple tokens share the same position ID remains theoretically unjustified. The authors claim it "works through training" but this lacks principled explanation and raises concerns about generalization to other models/tasks. (2) The task-dependent speedups (only 1.17x on math) significantly limit the method's practical value.

**Reviewer Scores:**

Reviewer Utbc (6): Would likely maintain score. The only reviewer above threshold but expressed willingness to reject.
Reviewer aASG (4): Might raise slightly to 6 or keep the same score of 4. Acknowledged some concerns addressed but also corrected authors' mischaracterization of PASTA, suggesting skepticism remains.
Reviewer MYWg (4): Would likely maintain score. Continued to express confusion about the position ID scheme even after rebuttal explanation, indicating the concern was not fully resolved.
Reviewer 9wUP (4): Would likely maintain score. Did not yet engage post-rebuttal.

---

### Decision · Program_Chairs · 2026-01-26

Reject